# FedSpeed: Larger Local Interval, Less Communication Round, and Higher Generalization Accuracy

**Yan Sun**
The University of Sydney
ysun9899@uni.sydney.edu.au

**Li Shen**[*]
JD Explore Academy
mathshenli@gmail.com

**Tiansheng Huang**
Georgia Institute of Technology
tianshenghuangscut@gmail.com

**Liang Ding**
JD Explore Academy
liangding.liam@gmail.com

**Dacheng Tao**
JD Explore Academy & The University of Sydney
dacheng.tao@gmail.com

## Abstract

Federated learning is an emerging distributed machine learning framework which jointly trains a global model via a large number of local devices with data privacy protections. Its performance suffers from the non-vanishing biases introduced by the local inconsistent optimal and the rugged client-drifts by the local over-fitting. In this paper, we propose a novel and practical method, FedSpeed, to alleviate the negative impacts posed by these problems. Concretely, FedSpeed applies the prox-correction term on the current local updates to efficiently reduce the biases introduced by the prox-term, a necessary regularizer to maintain the strong local consistency. Furthermore, FedSpeed merges the vanilla stochastic gradient with a perturbation computed from an extra gradient ascent step in the neighborhood, thereby alleviating the issue of local over-fitting. Our theoretical analysis indicates that the convergence rate is related to both the communication rounds $T$ and local intervals $K$ with a upper bound $\mathcal{O}(1/T)$ if setting a proper local interval. Moreover, we conduct extensive experiments on the real-world dataset to demonstrate the efficiency of our proposed FedSpeed, which performs significantly faster and achieves the state-of-the-art (SOTA) performance on the general FL experimental settings than several baselines including FedAvg, FedProx, FedCM, FedAdam, SCAFFOLD, FedDyn, FedADMM, etc.

## 1 Introduction

Since McMahan et al. (2017) proposed federated learning (FL), it has gradually evolved into an efficient paradigm for large-scale distributed training. Different from the traditional deep learning methods, FL allows multi local clients to jointly train a single global model without data sharing. However, FL is far from its maturity, as it still suffers from the considerable performance degradation over the heterogeneously distributed data, a very common setting in the practical application of FL.

We recognize the main culprit leading to the performance degradation of FL as *local inconsistency* and *local heterogeneous over-fitting*. Specifically, for canonical local-SGD-based FL method, e.g., FedAvg, the non-vanishing biases introduced by the local updates may eventually lead to inconsistent local solution. Then, the rugged client-drifts resulting from the local over-fitting into inconsistent local solutions may make the obtained global model degrading into the average of client's local parameters. The non-vanishing biases have been studied by several previous works Charles & Konečnỳ (2021);

---

[*]Li Shen is the corresponding author.

Malinovskiy et al. (2020) in different forms. The inconsistency due to the local heterogeneous data will compromise the global convergence during the training process. Eventually it leads to serious client-drifts which can be formulated as $\mathbf{x}^* \neq \sum_{i \in [m]} \mathbf{x}_i^* / m$. Larger data heterogeneity may enlarge the drifts, thereby degrading the practical training convergence rate and generalization performance.

In order to strengthen the local consistency during the local training process, and avoid the client-drifts resulting from the local over-fitting, we propose a novel and practical algorithm, dubbed as **FedSpeed**. Notably, FedSpeed incorporates two novel components to achieve SOTA performance. i) Firstly, FedSpeed inherits a penalized prox-term to force the local offset to be closer to the initial point at each communication round. However, recognized from Hanzely & Richtárik (2020); Khaled et al. (2019) that the prox-term between global and local solutions may introduce undesirable local training bias, we propose and utilize a prox-correction term to counteract the adverse impact. Indeed, in our theoretical analysis, the implication of the prox-correction term could be considered as a momentum-based term of the weighted local gradients. Via utilizing the historical gradient information, the bias brought by the prox-term can be effectively corrected. ii) Secondly, to avoid the rugged local over-fitting, FedSpeed incorporates a local gradient perturbation via merging the vanilla stochastic gradient with an extra gradient, which can be viewed as taking an extra gradient ascent step for each local update. Based on the analysis in Zhao et al. (2022); van der Hoeven (2020), we demonstrate that the gradient perturbation term could be approximated as adding a penalized squared $L2$-norm of the stochastic gradients to the original objective function, which can efficiently search for the flatten local minima Andriushchenko & Flammarion (2022) to prevent the local over-fitting problems.

We also provide the theoretical analysis of our proposed FedSpeed and further demonstrate that its convergence rate could be accelerated by setting an appropriate large local interval $K$. Explicitly, under the non-convex and smooth cases, FedSpeed with an extra gradient perturbation could achieve the fast convergence rate of $\mathcal{O}(1/T)$, which indicates that FedSpeed achieves a tighter upper bound with a proper local interval $K$ to converge, without applying a specific global learning rate or assuming the precision for the local solutions (Durmus et al., 2021; Wang et al., 2022). Extensive experiments are tested on CIFAR-10/100 and TinyImagenet dataset with a standard ResNet-18-GN network under the different heterogeneous settings, which shows that our proposed FedSpeed is significantly better than several baselines, e.g. for FedAvg, FedProx, FedCM, FedPD, SCAFFOLD, FedDyn, on both the stability to enlarge the local interval $K$ and the test generalization performance in the actual training.

To the end, we summarize the main contributions of this paper as follows:

- We propose a novel and practical federated optimization algorithm, **FedSpeed**, which applies a prox-correction term to significantly reduce the bias due to the local updates of the prox-term, and an extra gradient perturbation to efficiently avoid the local over-fitting, which achieves a fast convergence speed with large local steps and simultaneously maintains the high generalization.

- We provide the convergence rate upper bound under the non-convex and smooth cases and prove that FedSpeed could achieve a fast convergence rate of $\mathcal{O}(1/T)$ via enlarging the local training interval $K = \mathcal{O}(T)$ without any other harsh assumptions or the specific conditions required.

- Extensive experiments are conducted on the CIFAR-10/100 and TinyImagenet dataset to verify the performance of our proposed FedSpeed. To the best of our interests, both convergence speed and generalization performance could achieve the SOTA results under the general federated settings. FedSpeed could outperform other baselines and be more robust to enlarging the local interval.

## 2 RELATED WORK

McMahan et al. (2017) propose the federated framework with the properties of jointly training with several unbalance and non-iid local dataset via communicating with lower costs during the total training stage. The general FL optimization involves a local client training stage and a global server update operation Asad et al. (2020) and it has been proved to achieve a linear speedup property in Yang et al. (2021). With the fast development of the FL, a series of efficient optimization method are applied in the federated framework. Li et al. (2020b) and Kairouz et al. (2021) introduce a detailed overview in this field. There are still many difficulties to be solved in the practical scenarios, while in this paper we focus to highlight the two main challenges of the local inconsistent solution and client-drifts due to heterogeneous over-fitting, which are two acute limitations in the federated

optimization Li et al. (2020a); Yang et al. (2019); Konečný et al. (2016); Liu et al. (2022); Shi et al. (2023); Liu et al. (2023).

**Local consistency.** Sahu et al. (2018) study the non-vanishing biases of the inconsistent solution in the experiments and apply a prox-term regularization. FedProx utilizes the bounded local updates by penalizing parameters to provide a good guarantee of consistency. In Liang et al. (2019) they introduce the local gradient tracking to reduce the local inconsistency in the local SGD method. Charles & Konečný (2021); Malinovskiy et al. (2020) show that the local learning rate decay can balance the trade-off between the convergence rate and the local inconsistency with the rate of $\mathcal{O}(\eta_l(K-1))$. Furthermore, Wang et al. (2021; 2020b) through a simple counterexample to show that using adaptive optimizer or different hyper-parameters on local clients leads to an additional gaps. They propose a local correction technique to alleviate the biases. Wang et al. (2020a); Tan et al. (2022) consider the different local settings and prove that in the case of asynchronous aggregation, the inconsistency bias will no longer be eliminated by local learning rate decay. Haddadpour et al. (2021) compress the local offset and adopt a global correction to reduce the biases. Zhang et al. (2021) apply the primal dual method instead of the primal method to solve a series of sub-problems on the local clients and alternately updates the primal and dual variables which can achieve the fast convergence rate of $\mathcal{O}(\frac{1}{T})$ with the local solution precision assumption. Based on FedPD, Durmus et al. (2021) propose FedDyn as a variants via averaging all the dual variables (the average quantity can then be viewed as the global gradient) under the partial participation settings, which can also achieve the same $\mathcal{O}(\frac{1}{T})$ under the assumption that exact local solution can be found by the local optimizer. Wang et al. (2022); Gong et al. (2022) propose two other variants to apply different dual variable aggregation strategies under partial participation settings. These methods benefit from applying the prox-term Li et al. (2019); Chen & Chao (2020) or higher efficient optimization methods Bischoff et al. (2021); Yang et al. (2022) to control the local consistency.

**Client-drifts.** Karimireddy et al. (2020) firstly demonstrate the client-drifts for federated learning framework to indicate the negative impact on the global model when each local client over-fits to the local heterogeneous dataset. They propose SCAFFOLD via a variance reduction technique to mitigate this drifts. Yu et al. (2019) and Wang et al. (2019) introduce the momentum instead of the gradient to the local and global update respectively to improve the generalization performance. To maintain the property of consistency, Xu et al. (2021) propose a novel client-level momentum term to improve the local training process. Ozfatura et al. (2021) incorporate the client-level momentum with local momentum to further control the biases. In recent Gao et al. (2022); Kim et al. (2022), they propose a drift correction term as a penalized loss on the original local objective functions with a global gradient estimation. Chen et al. (2020) and Chen et al. (2021; 2022) focus on the adaptive method to alleviate the biases and improve the efficiency.

Our proposed FedSpeed inherits the prox-term at local update to guarantee the local consistency during local training. Different from the previous works, we adopt an extra prox-correction term to reduce the bias during the local training introduced by the update direction towards the last global model parameters. This ensures that the local update could be corrected towards the global minima. Furthermore, we incorporate a gradient perturbation update to enhance the generalization performance of the local model, which merges a gradient ascent step.

## 3 METHODOLOGY

In this part, we will introduce the preliminaries and our proposed method. We will explain the implicit meaning for each variables and demonstrate the FedSpeed algorithm inference in details.

**Notations and preliminary.** Let $m$ be the number of total clients. We denote $\mathcal{S}^t$ as the set of active clients at round $t$. $K$ is the number of local updates and $T$ is the communication rounds. $(\cdot)_{i,k}^t$ denotes variable $(\cdot)$ at $k$-th iteration of $t$-th round in the $i$-th client. $\mathbf{x}$ is the model parameters. $\mathbf{g}$ is the stochastic gradient computed by the sampled data. $\tilde{\mathbf{g}}$ is the weighted quasi-gradient computed as defined in Algorithm 1. $\hat{\mathbf{g}}$ is the prox-correction term. We denote $\langle \cdot, \cdot \rangle$ as the inner product for two vectors and $\| \cdot \|$ is the Euclidean norm of a vector. Other symbols are detailed at their references.

As the most FL frameworks, we consider to minimize the following finite-sum non-convex problem:

$$F(\mathbf{x}) = \frac{1}{m} \sum_{i=1}^{m} F_i(\mathbf{x}), \tag{1}$$

---

**Algorithm 1** FedSpeed Algorithm Framework

---

**Input:** model parameters $\mathbf{x}^0$, total communication rounds $T$, local gradient controller $\hat{\mathbf{g}}_i^{-1} = 0$, penalized weight $\lambda$.
**Output:** model parameters $\mathbf{x}^T$.
1: **for** $t = 0, 1, 2, \cdots, T-1$ **do**
2:     select active clients-set $\mathcal{S}^t$ at round $t$
3:     **for** client $i \in \mathcal{S}^t$ parallel **do**
4:         communicate $\mathbf{x}^t$ to local client $i$ and set $\mathbf{x}_{i,0}^t = \mathbf{x}^t$
5:         **for** $k = 0, 1, 2, \cdots, K-1$ **do**
6:             sample a minibatch $\varepsilon_{i,k}^t$ and do
7:             compute unbiased stochastic gradient: $\mathbf{g}_{i,k,1}^t = \nabla F_i(\mathbf{x}_{i,k}^t; \varepsilon_{i,k}^t)$
8:             update the extra step: $\breve{\mathbf{x}}_{i,k}^t = \mathbf{x}_{i,k}^t + \rho \mathbf{g}_{i,k,1}^t$
9:             compute unbiased stochastic gradient: $\mathbf{g}_{i,k,2}^t = \nabla F_i(\breve{\mathbf{x}}_{i,k}^t; \varepsilon_{i,k}^t)$
10:           compute quasi-gradient: $\tilde{\mathbf{g}}_{i,k}^t = (1-\alpha)\mathbf{g}_{i,k,1}^t + \alpha\mathbf{g}_{i,k,2}^t$
11:           update the gradient descent step: $\mathbf{x}_{i,k+1}^t = \mathbf{x}_{i,k}^t - \eta_l \left( \tilde{\mathbf{g}}_{i,k}^t - \hat{\mathbf{g}}_i^{t-1} + \frac{1}{\lambda}(\mathbf{x}_{i,k}^t - \mathbf{x}^t) \right)$
12:         **end for**
13:         $\hat{\mathbf{g}}_i^t = \hat{\mathbf{g}}_i^{t-1} - \frac{1}{\lambda}(\mathbf{x}_{i,K}^t - \mathbf{x}^t)$
14:         communicate $\hat{\mathbf{x}}_i^t = \mathbf{x}_{i,K}^t - \lambda\hat{\mathbf{g}}_i^t$ to the global server
15:     **end for**
16:     $\mathbf{x}^{t+1} = \frac{1}{S} \sum_{i \in \mathcal{S}^t} \hat{\mathbf{x}}_i^t$
17: **end for**

---

where $F \colon \mathbb{R}^d \to \mathbb{R}$, $F_i(\mathbf{x}) := \mathbb{E}_{\varepsilon_i \sim \mathcal{D}_i} F_i(\mathbf{x}, \varepsilon_i)$ is objective function in the client $i$, and $\varepsilon_i$ represents for the random data samples obeying the distribution $\mathcal{D}_i$. $m$ is the total number of clients. In FL, $\mathcal{D}_i$ may differ across the local clients, which may introduce the client drifts by the heterogeneous data.

## 3.1 FEDSPEED ALGORITHM

In this part, we will introduce our proposed method to alleviate the negative impact of the heterogeneous data and reduces the communication rounds. We are inspired by the dynamic regularization Durmus et al. (2021) for the local updates to eliminate the client drifts when $T$ approaches infinite.

Our proposed FeedSpeed is shown in Algorithm 1. At the beginning of each round $t$, a subset of clients $\mathcal{S}^t$ are required to participate in the current training process. The global server will communicate the parameters $\mathbf{x}^t$ to the active clients for local training. Each active local client performs three stages: (1) computing the unbiased stochastic gradient $\mathbf{g}_{i,k,1}^t = \nabla F_i(\mathbf{x}_{i,k}^t; \varepsilon_{i,k}^t)$ with a randomly sampled mini-batch data $\varepsilon_{i,k}^t$ and executing a gradient ascent step in the neighbourhood to approach $\breve{\mathbf{x}}_{i,k}^t$; (2) computing the unbiased stochastic gradient $\mathbf{g}_{i,k,2}^t$ with the same sampled mini-batch data in (1) at the $\breve{\mathbf{x}}_{i,k}^t$ and merging the $\mathbf{g}_{i,k,1}^t$ with $\mathbf{g}_{i,k,2}^t$ to introduce a basic perturbation to the vanilla descent direction; (3) executing the gradient descent step with the merged quasi-gradient $\tilde{\mathbf{g}}_{i,k}^t$, the prox-term $\|\mathbf{x}_{i,k}^t - \mathbf{x}^t\|^2$ and the local prox-correction term $\hat{\mathbf{g}}_i^{t-1}$. After $K$ iterations local training, prox-correction term $\hat{\mathbf{g}}_i^{t-1}$ will be updated as the weighted sum of the current local offset $(\mathbf{x}_{i,K}^t - \mathbf{x}_{i,0}^t)$ and the historical offsets momentum. Then we communicate the amended model parameters $\hat{\mathbf{x}}_i^t = \mathbf{x}_{i,K}^t - \lambda\hat{\mathbf{g}}_i^t$ to the global server for aggregation. On the global server, a simple average aggregation is applied to generate the current global model parameters $\mathbf{x}^{t+1}$ at round $t$.

**Prox-correction term.** In the general optimization, the prox-term $\|\mathbf{x}_{i,k}^t - \mathbf{x}^t\|^2$ is a penalized term for solving the non-smooth problems and it contributes to strengthen the local consistency in the FL framework by introducing a penalized direction in the local updates as proposed in Sahu et al. (2018). However, as discussed in Hanzely & Richtárik (2020), it simply performs as a balance between the local and global solutions, and there still exists the non-vanishing inconsistent biases among the local solutions, i.e., the local solutions are still largely deviated from each other, implying that local inconsistency is still not eliminated, which limits the efficiency of the federated learning framework.

To further strengthen the local consistency, we utilize a prox-correction term $\hat{\mathbf{g}}_i^t$ which could be considered as a previous local offset momentum. According to the local update, we combine the $\mathbf{x}_{i,k-1}^t$ term in the prox term and the local state, setting the weight as $(1 - \frac{\eta_l}{\lambda})$ multiplied to the basic

local state. As shown in the local update in Algorithm 1 (Line.11), for $\forall\ \mathbf{x} \in \mathbb{R}^d$ we have:

$$\mathbf{x}_{i,K}^t - \mathbf{x}^t = -\gamma\lambda \sum_{k=0}^{K-1} \frac{\gamma_k}{\gamma} \tilde{\mathbf{g}}_{i,k}^t + \gamma\lambda\hat{\mathbf{g}}_i^{t-1}, \qquad (2)$$

where $\sum_{k=0}^{K-1} \gamma_k = \sum_{k=0}^{K-1} \frac{\eta_l}{\lambda}\left(1 - \frac{\eta_l}{\lambda}\right)^{K-1-k} = \gamma$. Proof details can be referred to the **Appendix**.

Firstly let $\hat{\mathbf{g}}_i^{-1} = \mathbf{0}$, Equation (2) indicates that the local offset will be transferred to a exponential average of previous local gradients when applying the prox-term, and the updated formation of the local offset is independent of the local learning rate $\eta_l$. This is different from the vanilla SGD-based methods, e.g. FedAvg, which treats all local updates fairly. $\gamma_k$ changes the importance of the historical gradients. As $K$ increases, previous updates will be weakened by exponential decay significantly for $\eta_l < \lambda$. Thus, we apply the prox-correction term to balance the local offset. According to the iterative formula for $\hat{\mathbf{g}}_i^t$ (Line.13 in Algorithm 1) and the equation (2), we can rewrite this update as:

$$\hat{\mathbf{g}}_i^t = (1-\gamma)\hat{\mathbf{g}}_i^{t-1} + \gamma\left(\sum_{k=0}^{K-1} \frac{\gamma_k}{\gamma} \tilde{\mathbf{g}}_{i,k}^t\right), \qquad (3)$$

where $\gamma$ and $\gamma_k$ is defined the same as in Equation (2). Proof details can be referred to the **Appendix**.

Note that $\hat{\mathbf{g}}_i^t$ performs as a momentum term of the historical local updates before round $t$, which can be considered as a estimation of the local offset at round $t$. At each local iteration $k$ of round $t$, $\hat{\mathbf{g}}_i^{t-1}$ provides a correction for the local update to balance the impact of the prox-term to enhance the contribution of those descent steps executed firstly at each local stages. It should be noted that $\hat{\mathbf{g}}_i^{t-1}$ is different from the global momentum term mentioned in Wang et al. (2019) which aggregates the average local updates to improve the generalization performance. After the local training, it updates the current information. Then we subtract the current $\hat{\mathbf{g}}_i^t$ from the local models $\mathbf{x}_{i,K}^t$ to counteract the influence in the local stages. Finally it sends the post-processed parameters $\hat{\mathbf{x}}_{i,K}^t$ to the global server.

**Gradient perturbation.** Gradient perturbations (Foret et al., 2020a; Mi et al.; Zhao et al., 2022; Zhong et al., 2022) significantly improves generalization for deep models. An extra gradient ascent in the neighbourhood can effectively express the curvature near the current parameters. Referring to the analysis in Zhao et al. (2022), we show that the quasi-gradient $\tilde{\mathbf{g}}$, which merges the extra ascent step gradient and the vanilla gradient, could be approximated as penalizing a square term of the $L2$-norm of the gradient on the original function. On each local client to solve the stationary point of $\min_{\mathbf{x}}\{F_i(\mathbf{x}) + \beta\|\nabla F_i(\mathbf{x})\|^2\}$ can search for a flat minima. Flatten loss landscapes will further mitigate the local inconsistency due to the averaging aggregation on the global server on heterogeneous dataset. Detailed discussions can be referred to the **Appendix**.

## 4 CONVERGENCE ANALYSIS

In this part we will demonstrate the theoretical analysis of our proposed FedSpeed and illustrate the convergence guarantees under the specific hyperparameters. Due to the space limitations, more details could be referred to the **Appendix**. Some standard assumptions are stated as follows.

**Assumption 4.1** (*L-Smoothness*) *For the non-convex function $F_i$ holds the property of smoothness for all $i \in [m]$, i.e., $\|\nabla F_i(\mathbf{x}) - \nabla F_i(\mathbf{y})\| \le L\|\mathbf{x} - \mathbf{y}\|$, for all $\mathbf{x}, \mathbf{y} \in \mathbb{R}^d$.*

**Assumption 4.2** (*Bounded Stochastic Gradient*) *The stochastic gradient $\mathbf{g}_{i,k}^t = \nabla F_i(\mathbf{x}_{i,k}^t, \varepsilon_{i,k}^t)$ with the randomly sampled data $\varepsilon_{i,k}^t$ on the local client $i$ is an unbiased estimator of $\nabla F_i$ with bounded variance, i.e., $\mathbb{E}[\mathbf{g}_{i,k}^t] = \nabla F_i(\mathbf{x}_{i,k}^t)$ and $\mathbb{E}\|\mathbf{g}_{i,k}^t - \nabla F_i(\mathbf{x}_{i,k}^t)\|^2 \le \sigma_l^2$, for all $\mathbf{x}_{i,k}^t \in \mathbb{R}^d$.*

**Assumption 4.3** (*Bounded Heterogeneity*) *The dissimilarity of the dataset among the local clients is bounded by the local and global gradients, i.e., $\mathbb{E}\|\nabla F_i(\mathbf{x}) - \nabla F(\mathbf{x})\|^2 \le \sigma_g^2$, for all $\mathbf{x} \in \mathbb{R}^d$.*

Assumption 4.1 guarantees a Lipschitz continuity and Assumption 4.2 guarantees the stochastic gradient is bounded by zero mean and constant variance. Assumption 4.3 is the heterogeneity bound for the non-iid dataset, which is widely used in many previous works (Reddi et al., 2020; Yang et al.,

2021; Xu et al., 2021; Wang et al., 2021; Karimi et al., 2021). Our theoretical analysis depends on the above assumptions to explore the comprehensive properties in the local training process.

**Proof sketch.** To express the essential insights in the updates of the Algorithm 1, we introduce two auxiliary sequences. Considering the $\mathbf{u}^t = \frac{1}{m} \sum_{i \in [m]} \mathbf{x}^t_{i,K}$ as the mean averaged parameters of the last iterations in the local training among the local clients. Based on $\{\mathbf{u}^t\}$, we introduce the auxiliary sequences $\{\mathbf{z}^t = \mathbf{u}^t + \frac{1-\gamma}{\gamma}(\mathbf{u}^t - \mathbf{u}^{t-1})\}_{t>0}$. Combining the local update and the Equation (3):

$$\mathbf{u}^{t+1} = \mathbf{u}^t - \lambda \frac{1}{m} \sum_{i \in [m]} \sum_{k=0}^{K-1} \frac{\gamma_k}{\gamma} \left( \gamma \tilde{\mathbf{g}}^t_{i,k} + (1-\gamma) \hat{\mathbf{g}}^{t-1}_i \right). \tag{4}$$

If we introduce $K$ virtual states $\mathbf{u}^t_{i,k}$, it could be considered as a momentum-based update of the prox-correction term $\hat{\mathbf{g}}^{t-1}_i$ with the coefficient $\gamma$. And the prox-correction term $\hat{\mathbf{g}}^t_i = -\frac{1}{\lambda} \left( \mathbf{u}^t_{i,K} - \mathbf{u}^t_{i,0} \right)$, which implies the global update direction in the local training process. And $\mathbf{z}^t$ updates as:

$$\mathbf{z}^{t+1} = \mathbf{z}^t - \lambda \frac{1}{m} \sum_{i \in [m]} \sum_{k=0}^{K-1} \frac{\gamma_k}{\gamma} \tilde{\mathbf{g}}^t_{i,k}, \tag{5}$$

Detailed proofs can be referred in the **Appendix**. After mapping $\mathbf{x}^t$ to $\mathbf{u}^t$, the local update could be considered as a client momentum-like method with a normalized weight parameterized by $\gamma_k$. Further, after mapping $\mathbf{u}^t$ to $\mathbf{z}^t$, the entire update process will be simplified to a SGD-type method with the quasi-gradients $\tilde{\mathbf{g}}$. $\mathbf{z}^t$ contains the penalized prox-term in the total local training stage. Though a prox-correction term is applied to eliminate the local biases, $\mathbf{x}^t$ maintains to be beneficial from the update of penalizing the prox-term. The prox-correction term plays the role as exponential average of the global offset. Then we introduce our proof of the convergence rate for the FedSpeed algorithm:

**Theorem 4.4** *Under the Assumptions 4.1-4.3, when the perturbation learning rate satisfies $\rho \le \frac{1}{\sqrt{6}\alpha L}$, and the local learning rate satisfies $\eta_l \le \min\{\frac{1}{32\sqrt{3}KL}, 2\lambda\}$, and the local interval satisfies $K \ge \lambda/\eta_l$, let $\kappa = \frac{1}{2} - 3\alpha^2 L^2 \rho^2 - 1536 \eta_l^2 L^2 K$ is a positive constant with selecting the proper $\eta_l$ and $\rho$, the auxiliary sequence $\mathbf{z}^t$ in Equation (5) generated by executing the Algorithm 1 satisfies:*

$$\frac{1}{T} \sum_{t=1}^{T-1} \mathbb{E}\|\nabla F(\mathbf{z}^t)\|^2 \le \frac{2(F(\mathbf{z}^1) - F^*)}{\lambda \kappa T} + \frac{64\eta_l L^2 K}{\kappa m T} \sum_{i \in [m]} \mathbb{E}\|\hat{\mathbf{g}}^0_i\|^2 + \frac{32\lambda^2 L^2}{\kappa T} \mathbb{E}_t \|\frac{1}{m} \sum_{i \in [m]} \hat{\mathbf{g}}^0_i\|^2 + \Phi,$$
$$\tag{6}$$

*where $F$ is a non-convex objective function $F^*$ is the optimal of $F$. The term $\Phi$ is:*

$$\Phi = \frac{1}{\kappa} \left( 32\lambda\eta_l^2 L^2 K(16\sigma_g^2 + \sigma_l^2) + \lambda\alpha^2 L^2 \rho^2 (3\sigma_g^2 + \sigma_l^2) \right), \tag{7}$$

*where $\alpha$ is the perturbation weight. More proof details can be referred to the **Appendix**.*

**Corollary 4.5** *Let $\rho = \mathcal{O}(1/\sqrt{T})$ with the upper bound of $\rho \le 1/\sqrt{6}\alpha L$, and let $\eta_l = \mathcal{O}(1/K)$ with the lower bound of $\eta_l \ge \lambda/K$, when the local interval $K$ is long enough with $K = \mathcal{O}(T)$, the proposed FedSpeed achieves a fast convergence rate of $\mathcal{O}(1/T)$.*

**Remark 4.6** *Compared with the other prox-based works, e.g. for (Durmus et al., 2021; Wang et al., 2022; Gong et al., 2022), their proofs rely on the harsh assumption that local client must approach an exact stationary point or $\epsilon$-inexact stationary point in the local training per round. It cannot be strictly satisfied in the practical federated learning framework with the current theoretical analysis of the last iteration point on the non-convex case. We relax this assumption through enlarging the local interval and prove that federated prox-based methods can also achieve the convergence of $\mathcal{O}(1/T)$.*

**Remark 4.7** *Compared with the other current methods, FedSpeed can improve the convergence rate by increasing the local interval $K$, which is a good property for the practical federated learning framework. For the analysis of FedAvg (Yang et al., 2021), under the same assumptions, it achieves $\mathcal{O}(1/\sqrt{SKT} + K/T)$ which restricts the value of $K$ to not exceed the order of $T$. Karimireddy et al. (2020) contribute the convergence as $\mathcal{O}(1/\sqrt{SKT})$ under the constant local interval, and (Reddi et al., 2020) proves the same convergence under the strict coordinated bounded variance assumption for the global full gradient in the FedAdam. Our experiments also verify this characteristic in Section 5.3. Most current algorithms are affected by increasing $K$ in the training while FedSpeed shows the good stability under the enlarged local intervals and shrunk communication rounds.*

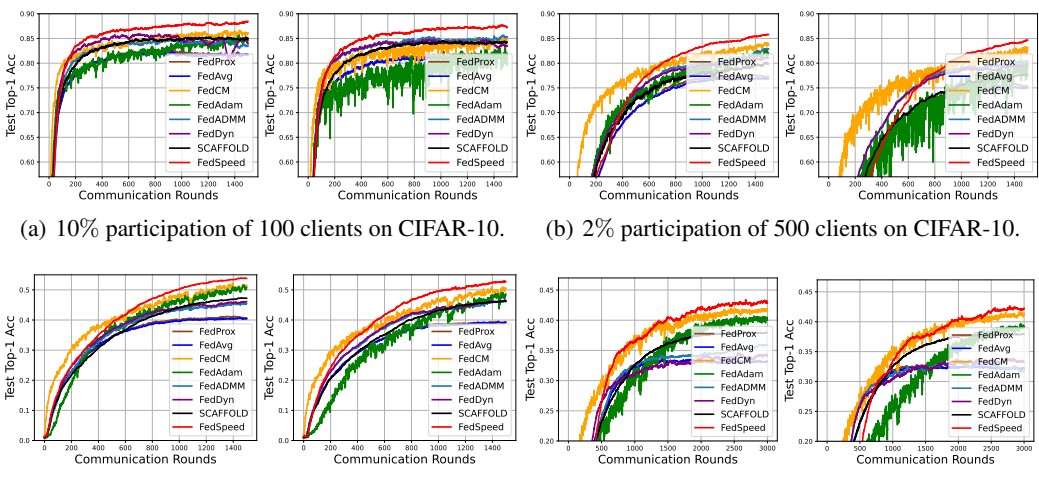

(a) 10% participation of 100 clients on CIFAR-10.   (b) 2% participation of 500 clients on CIFAR-10.

(c) 2% participation of 500 clients on CIFAR-100.   (d) 2% participation of 500 clients on TinyImagenet.

Figure 1: The top-1 accuracy in communication rounds of all compared methods on CIFAR-10/100 and TinyImagenet. Communication rounds are set as 1500 for CIFAR-10/100, 3000 for TinyImagenet. In each group, the left shows the performance on IID dataset while the right shows the performance on the non-IID dataset, which are split by setting heterogeneity weight of the Dirichlet as 0.6.

## 5 EXPERIMENTS

In this part, we firstly introduce our experimental setups. We present the convergence and generalization performance in Section 5.2, and study ablation experiments in Section 5.3.

### 5.1 SETUP

**Dataset and backbones.** We test the experiments on CIFAR-10, CIFAR-100 Krizhevsky et al. (2009) and TinyImagenet. Due to the space limitations we introduce these datasets in the **Appendix**. We follow the Hsu et al. (2019) to introduce the heterogeneity via splitting the total dataset by sampling the label ratios from the Dirichlet distribution. We train and test the performance on the standard ResNet-18 He et al. (2016) backbone with the $7\times7$ filter size in the first convolution layer with BN-layers replaced by GN Wu & He (2018); Hsieh et al. (2020) to avoid the invalid aggregation.

**Implementation details.** We select each hyper-parameters within the appropriate range and present the combinations under the best performance. To fairly compare these baseline methods, we fix the most hyper-parameters for all methods under the same setting. For the 10% participation of total 100 clients training, we set the local learning rate as 0.1 initially and set the global learning rate as 1.0 for all methods except for FedAdam which applies 0.1 on global server. The learning rate decay is set as multiplying 0.998 per communication round except for FedDyn, FedADMM and FedSpeed which apply 0.9995. Each active local client trains 5 epochs with batchsize 50. Weight decay is set as $1e$-3 for all methods. The weight for the prox-term in FedProx, FedDyn, FedADMM and FedSpeed is set as 0.1. For the 2% participation, the learning rate decay is adjusted to 0.9998 for FedDyn and FedSpeed. Each active client trains 2 epochs with batchsize 20. The weight for the prox-term is set as 0.001. The other hyper-parameters specific to each method will be introduced in the **Appendix**.

**Baselines.** We compare several classical and efficient methods with the proposed FedSpeed in our experiments, which focus on the local consistency and client-drifts, including FedAvg McMahan et al. (2017), FedAdam Reddi et al. (2020), SCAFFOLD Karimireddy et al. (2020), FedCM Xu et al. (2021), FedProx Sahu et al. (2018), FedDyn Durmus et al. (2021) and FedADMM Wang et al. (2022). FedAdam applies adaptive optimizer to improve the performance on the global updates. SCAFFOLD and FedCM utilize the global gradient estimation to correct the local updates. FedProx introduces the prox-term to alleviate the local inconsistency. FedDyn and FedADMM both employ the different variants of the primal-dual method to reduce the local inconsistency. Due to the limited space, more detailed description and discussions on these compared baselines are placed in the **Appendix**.

Table 1: Test accuracy (%) on the CIFAR-10/100 and TinyImagenet under the 2% participation of 500 clients with IID and non-IID dataset. The heterogeneity is applied as Dirichlet-0.6 (**DIR.**).

| Method | CIFAR-10 | | CIFAR-100 | | TinyImagenet | |
|---|---|---|---|---|---|---|
| | IID. | DIR. | IID. | DIR. | IID. | DIR. |
| FedAvg | 77.01 | 75.21 | 40.68 | 39.33 | 33.58 | 32.71 |
| FedProx | 77.68 | 75.97 | 41.29 | 39.69 | 33.71 | 32.78 |
| FedAdam | 82.92 | 80.55 | 51.65 | 49.29 | 40.85 | 39.71 |
| SCAFFOLD | 80.11 | 77.71 | 47.38 | 46.33 | 38.03 | 37.54 |
| FedCM | 84.20 | 83.48 | 52.35 | 50.98 | 41.90 | 41.67 |
| FedDyn | 83.36 | 80.57 | 46.18 | 46.60 | 34.69 | 33.92 |
| FedADMM | 81.29 | 79.71 | 45.51 | 46.65 | 36.03 | 33.83 |
| **FedSpeed** | **85.80** | **84.79** | **53.93** | **52.88** | **43.38** | **42.75** |

## 5.2 EXPERIMENTS

**CIFAR-10.** Our proposed FedSpeed is robust to different participation cases. Figure 1 (a) shows the results of 10% participation of total 100 clients. For the IID splits, FedSpeed achieves 6.1% ahead of FedAvg as 88.5%. FedDyn suffers the instability when learning rate is small, which is the similar phenomenon as mentioned in Xu et al. (2021). When introducing the heterogeneity, FedAdam suffers from the increasing variance obviously with the accuracy dropping from 85.7% to 83.2%. Figure 1 (b) shows the impact from reducing the participation. FedAdam is lightly affected by this change while the performance degradation of SCAFFOLD is significant which drops from 85.3% to 80.1%.

**CIFAR-100 & TinyImagenet.** As shown in Figure 1 (c) and (d), the performance of FedSpeed on the CIFAR-100 and TinyImagenet with low participating setting performs robustly and achieves approximately 1.6% and 1.8% improvement ahead of the FedCM respectively. As the participation is too low, the impact from the heterogeneous data becomes weak gradually with a similar test accuracy. SCAFFOLD is still greatly affected by a low participation ratio, which drops about 3.3% lower than FedAdam. FedCM converges fast at the beginning of the training stage due to the benefits from strong consistency limitations. FedSpeed adopts to update the prox-correction term and converges faster with its estimation within several rounds and then FedSpeed outperforms other methods.

Table 1 shows the accuracy under the low participation ratio equals to 2%. Our proposed FedSpeed outperforms on each dataset on both IID and non-IID settings. Table 1 shows the accuracy under the low participation ratio equals to 2%. Our proposed FedSpeed outperforms on each dataset on both IID and non-IID settings. We observe the similar results as mentioned in Reddi et al. (2020); Xu et al. (2021). FedAdam and FedCM could maintain the low consistency in the local training stage with a robust results to achieve better performance than others. While FedDyn is affected greatly by the number of training samples in the dataset, which is sensitive to the partial participation ratios.

**Large local interval for the prox-term.** From the IID case to the non-IID case, the heterogeneous dataset introduces the local inconsistency and leads to the severe client-drifts problem. Almost all the baselines suffer from the performance degradation. High local consistency usually supports for a large interval as for their bounded updates and limited offsets. Applying prox-term guarantees the local consistency, but it also has an negative impact on the local training towards the target of weighted local optimal and global server model. FedDyn and FedADMM succeed to apply the primal-dual method to alleviate this influence as they change the local objective function whose target is reformed by a dual variable. These method can mitigate the local offsets caused by the prox-term and they improve about 3% ahead of the FedProx on CIFAR-10. However, the primal-dual method requires a local $\epsilon$-close solution. In the non-convex optimization it is difficult to determine the selection of local training interval $K$ under this requirement. Though Durmus et al. (2021) claim that 5 local epochs are approximately enough for the $\epsilon$-close solution, there is still an unpredictable local biases.

FedSpeed directly applies a prox-correction term to update $K$ epochs and avoids the requirement for the precision of local solution. This ensures that the local optimization stage does not introduce the bias due to the error of the inexact solution. Moreover, the extra ascent step can efficiently improve the performance of local model parameters. Thus, the proposed FedSpeed can improve 3% than FedDyn and FedADMM and achieve the comparable performance as training on the IID dataset.

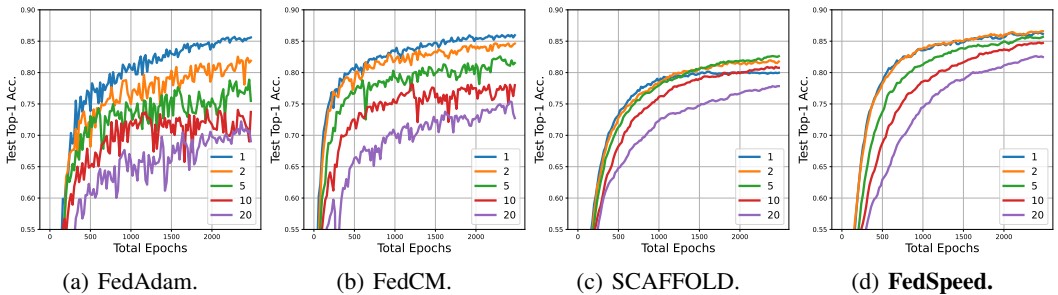



(a) FedAdam.     (b) FedCM.     (c) SCAFFOLD.     (d) **FedSpeed.**



Figure 2: Performance of FedAdam, FedCM, SCAFFOLD and FedSpeed with local epochs $E = 1, 2, 5, 10, 20$ on the $10\%$ participation case of total 100 clients on CIFAR-10. We fix $T \times E = 2500$ as the equaled total training epochs to illustrate the performance of increasing $E$ and decreasing $T$.

An interesting experimental phenomenon is that the performance of SCAFFOLD gradually degrades under the low participation ratio. It should be noticed that under the $10\%$ participation case, SCAFFOLD performs as well as the FedCM. It benefits from applying a global gradient estimation to correct the local updates, which can weaken the client-drifts by a quasi gradient towards to the global optimal. Actually the estimation variance is related to the participation ratio, which means that their efficiencies rely on the enough number of clients. When the participation ratio decreases to be extremely low, their performance will also be greatly affected by the huge biases in the local training.

### 5.3 HYPERPARAMETERS SENSITIVITY

**Local interval $K$.** To explore the acceleration on $T$ by applying a large interval $K$, we fix the total training epochs $E$. It should be noted that $K$ represents for the iteration and $E$ represents for the epoch. A larger local interval can be applied to accelerate the convergence in many previous works theoretically, e.g. for SCAFFOLD and FedAdam, while empirical studies are usually unsatisfactory. As shown in Figure 2, in the FedAdam and FedCM, when $K$ increases from 1 to 20, the accuracy drops about $13.7\%$ and $10.6\%$ respectively. SCAFFOLD is affected lightly while its performance is much lower. In Figure 2 (d), FedSpeed applies the larger $E$ to accelerate the communication rounds $T$ both on theoretical proofs and empirical results, which stabilizes to swing within $3.8\%$ lightly.

**Learning rate $\rho$ for gradient perturbation.** In the simple analysis, $\rho$ can be selected as a proper value which has no impact on the convergence complexity. By noticing that if $\alpha \neq 0$, $\rho$ could be selected irrelevant to $\eta_l$. To achieve a better

Table 2: Performance of different $\rho_0$ with $\alpha = 1$.

| $\rho_0$ | 0 | 0.01 | 0.05 | **0.1** | 0.2 |
|---|---|---|---|---|---|
| Acc. | 83.97 | 84.6 | 85.38 | **85.72** | 84.35 |

performance, we apply the ascent learning rate $\rho = \rho_0 / \|\nabla F_i\|$ to in the experiments, where $\rho_0$ is a constant value selected from the Table 2. $\rho$ is consistent with the sharpness aware minimization Foret et al. (2020b) which can search for a flat local minimal. Table 2 shows the performance of utilizing the different $\rho_0$ on CIFAR-10 by 500 communication rounds under the $10\%$ participation of total 100 clients setting. Due to the space limitations more details could be referred to the **Appendix**.

## 6 CONCLUSION

In this paper, we propose a novel and practical federated method FedSpeed which applies a prox-correction term to neutralize the bias due to prox-term in each local training stage and utilizes a perturbation gradient weighted by an extra gradient ascent step to improve the local generalization performance. We provide the theoretical analysis to guarantee its convergence and prove that FedSpeed benefits from a larger local interval $K$ to achieve a fast convergence rate of $\mathcal{O}(1/T)$ without any other harsh assumptions. We also conduct extensive experiments to highlight the significant improvement and efficiency of our proposed FedSpeed, which is consistent with the properties of our analysis. This work inspires the FL framework design to focus on the local consistency and local higher generalization performance to implement the high-efficient method to federated learning.

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

## A  GRADIENT PERTURBATION

### A.1  UNDERSTANDING OF GRADIENT PERTURBATION

We propose the gradient perturbation in the local training stage instead of the traditional stochastic gradient, which merges an extra gradient ascent step to the vanilla gradient by a hyper-parameter $\alpha$. While its ascent step usually approximates the worst point in the neighbourhood. This has been studied in many previous works, e.g. for the form of extra gradient and the sharpness aware minimization. In our studies, we perform the extra gradient ascent step instead of the descent step in extra gradient method. It also could be considered as a variant of the sharpness aware minimization method via weighted averaging the ascent step gradient and the vanilla gradient, instead of the normalized gradient. Here we illustrate the implicit of this quasi-gradient $\tilde{g}$ in our proposed FedSpeed and explain the positive efficiency for the local training from the perspective of objective functions.

Firstly we consider to minimize the non-convex problem $\mathcal{L}_p(\mathbf{x})$. To approach the stationary point of $\mathcal{L}_p$, we can simply introduce a penalized gradient term as a extra loss in $\mathcal{L}_p$, which is to solve the problem $\min_{\mathbf{x}}\{\mathcal{L}(\mathbf{x}) \triangleq \mathcal{L}_p(\mathbf{x}) + \frac{\beta}{2}\|\nabla\mathcal{L}_p(\mathbf{x})\|^2\}$. The final optimization target is consistent with the vanilla target, while penalizing gradient term can approach a flatten minimal empirically. We compute the gradient form as follows:

$$\nabla\mathcal{L}(\mathbf{x}) = \nabla\mathcal{L}_p(\mathbf{x}) + \frac{\beta}{2}\nabla\|\nabla\mathcal{L}_p(\mathbf{x})\|^2 = \nabla\mathcal{L}_p(\mathbf{x}) + \beta\nabla^2\mathcal{L}_p(\mathbf{x}) \cdot \nabla\mathcal{L}_p(\mathbf{x}). \quad (8)$$

The update in Equation (8) contains second-order Hessian information, which involves a huge amount of parameters for calculation. To further simplify the updates, we consider an approximation for the gradient form. We expand the function $\mathcal{L}_p$ via Taylor expansion as:

$$\mathcal{L}_p(\mathbf{x} + \Delta) = \mathcal{L}_p(\mathbf{x}) + \nabla\mathcal{L}_p(\mathbf{x})\Delta + \frac{1}{2}\Delta^T\nabla^2\mathcal{L}_p(\mathbf{x})\Delta + \mathcal{R}_\Delta,$$

where $\mathcal{R}_\Delta = \mathcal{O}(\|\Delta\|^2)$ is the infinitesimal to $\|\Delta\|^2$, which is directly omitted in our approximation. Thus we have the gradient form on $\Delta$ as:

$$\nabla\mathcal{L}_p(\mathbf{x} + \Delta) \approx \nabla\mathcal{L}_p(\mathbf{x}) + \nabla^2\mathcal{L}_p(\mathbf{x})\Delta.$$

$\mathcal{R}_\Delta$ is relevant to $\Delta$. We set the $\Delta = \rho\nabla\mathcal{L}_p(\mathbf{x})$ and then we have:

$$\nabla^2\mathcal{L}_p(\mathbf{x})\nabla\mathcal{L}_p(\mathbf{x}) \approx \frac{1}{\rho}\big(\nabla\mathcal{L}_p(\mathbf{x} + \rho\nabla\mathcal{L}_p(\mathbf{x})) - \nabla\mathcal{L}_p(\mathbf{x})\big). \quad (9)$$

Thus we connect Equation (8) and Equation (9), we have:

$$\begin{aligned}
\nabla\mathcal{L}(\mathbf{x}) &= \nabla\mathcal{L}_p(\mathbf{x}) + \beta\nabla^2\mathcal{L}_p(\mathbf{x}) \cdot \nabla\mathcal{L}_p(\mathbf{x}) \\
&\approx \nabla\mathcal{L}_p(\mathbf{x}) + \frac{\beta}{\rho}\big(\nabla\mathcal{L}_p(\mathbf{x} + \rho\nabla\mathcal{L}_p(\mathbf{x})) - \nabla\mathcal{L}_p(\mathbf{x})\big) \\
&= \big(1 - \frac{\beta}{\rho}\big)\nabla\mathcal{L}_p(\mathbf{x}) + \frac{\beta}{\rho}\nabla\mathcal{L}_p\big(\mathbf{x} + \rho\nabla\mathcal{L}_p(\mathbf{x})\big) \\
&= (1 - \alpha)\nabla\mathcal{L}_p(\mathbf{x}) + \alpha\nabla\mathcal{L}_p\big(\mathbf{x} + \rho\nabla\mathcal{L}_p(\mathbf{x})\big).
\end{aligned}$$

**Here we can see that the balance weight $\alpha$ in our proposed method is actually the ratio of the gradient penalized weight $\beta$ and the gradient ascent step size $\rho$.** To fix the step size $\rho$, increasing $\alpha$

means increasing the gradient penalized weight $\beta$, which facilitates searching for a flatten stationary point to improve the generalization performance. While the second term of $\nabla \mathcal{L}(\mathbf{x})$ can not be directly computed for its nested form, we approximate the second term with the chain rule as follows:

$$\nabla \mathcal{L}_p\big(\mathbf{x} + \rho \nabla \mathcal{L}_p(\mathbf{x})\big) \approx \nabla \mathcal{L}_p(\theta)|_{\theta=\mathbf{x}+\rho\nabla\mathcal{L}_p(\mathbf{x})}.$$

Finally we have:

$$\nabla \mathcal{L}(\mathbf{x}) \approx (1 - \alpha)\nabla \mathcal{L}_p(\mathbf{x}) + \alpha \nabla \mathcal{L}_p(\theta)|_{\theta=\mathbf{x}+\rho\nabla\mathcal{L}_p(\mathbf{x})}. \tag{10}$$

The Equation (10) provides an understanding for the weighted quasi gradient $\tilde{\mathbf{g}}$ on the local training stage in our proposed FedSpeed. We select an appropriate $0 \leq \beta \leq \rho$ to satisfy the update of perturbation gradient. It executes a gradient ascent step firstly with the step size $\rho$ to $\check{\mathbf{x}}$. Then it generates the stochastic gradient by the same sampled mini-batch data as the ascent step at $\check{\mathbf{x}}$. The quasi-gradient is merged as Equation (10) to execute the gradient descent step.

This is just a simple approximation for the gradient perturbation to help for understanding the implicit of the quasi-gradient and its performance in the training stage. Actually the error of the approximation depends a lot on $\rho$. The smaller $\rho$, the higher the accuracy of this estimation, but the smaller $\rho$, the less efficient the optimizer performs. Similar understanding can be referred in the (Qu et al., 2022; Caldarola et al., 2022; Andriushchenko & Flammarion, 2022).

# B EXPERIMENTS

## B.1 SETUPS

Table 3: Dataset introductions.

| Dataset | Training Data | Test Data | Class | Size |
|---------|---------------|-----------|-------|------|
| CIFAR-10 | 50,000 | 10,000 | 10 | $3\times32\times32$ |
| CIFAR-100 | 50,000 | 10,000 | 100 | $3\times32\times32$ |
| TinyImagenet | 100,000 | 10,000 | 200 | $3\times64\times64$ |

**Dataset and Backbones.** Extensive experiments are tested on CIFAR-10/100 dataset. We test on the two different settings as 10% participation of total 100 clients and 2% participation of total 500 clients. CIFAR-10 dataset contains 50,000 training data and 10,000 test data in 10 classes. Each data sample is a $3\times32\times32$ color image. CIFAR-100 Krizhevsky et al. (2009) includes 50,000 training data and 10,000 test data in 100 classes as 500 training samples per class. TinyImagenet involves 100,000 training images and 10,000 test images in 200 classes for $3\times64\times64$ color images, as shown in Table 3. To fairly compare with the other baselines, we train and test the performance on the standard ResNet-18 He et al. (2016) backbone with the $7\times7$ filter size in the first convolution layer as implemented in the previous works, e.g. for Karimireddy et al. (2020); Durmus et al. (2021); Xu et al. (2021). We follow the Hsieh et al. (2020) to replace the batch normalization layer with group normalization layer Wu & He (2018), which can be aggregated directly by averaging. These are all common setups in many previous works.

**Dataset Partitions.** To fairly compare with the other baselines, we follow the Hsu et al. (2019) to introduce the heterogeneity via splitting the total dataset by sampling the label ratios from the Dirichlet distribution. An additional parameter is used to control the level of the heterogeneity of the entire data partition. In order to visualize the distribution of heterogeneous data, we make the heat maps of the label distribution in different dataset, as shown in Figure 3. Since the heat map of 500 clients cannot be displayed normally, we show 100 clients case. It could be seen that for heterogeneity weight equals to 0.6, about 10% to 20% of the categories dominate on each client, which is white block in the Figure 3. The IID dataset is totally averaged in each client.

**Data Argumentation.** For CIFAR-10/100, we follow the implementation in the Karimireddy et al. (2020); Durmus et al. (2021) to normalize the pixel value within a specific mean and std value in our code, which are [0.491, 0.482, 0.447] for mean, [0.247, 0.243, 0.262] for std and [0.5071, 0.4867, 0.4408] for mean, [0.2675, 0.2565, 0.2761] for std. We randomly flip the training samples and randomly crop the images enlarged with the padding equal to 4. For TinyImagenet, the same argumentation is applied except for the padding equal to 8.

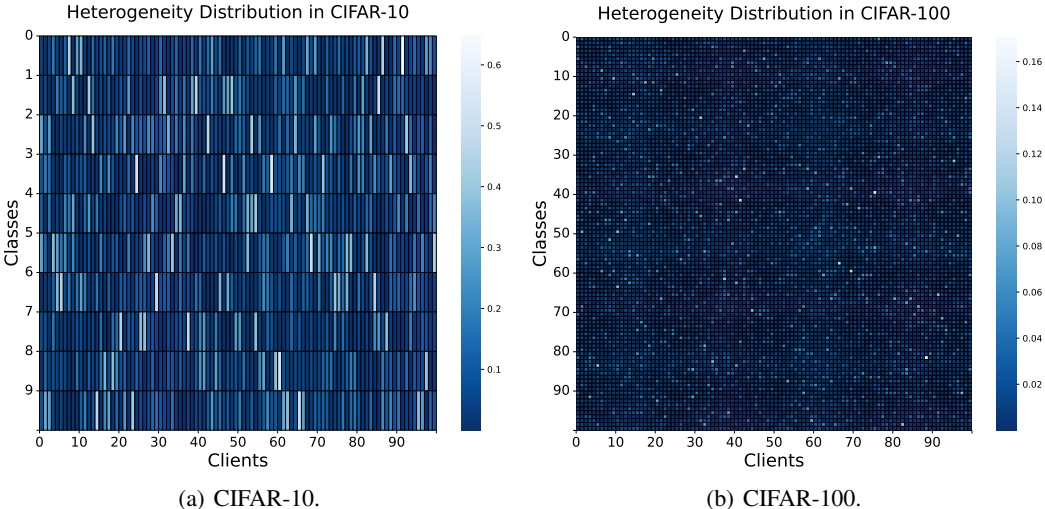

(a) CIFAR-10.  (b) CIFAR-100.

Figure 3: Heat maps for different dataset under heterogeneity weight equals to 0.6 for Dirichlet distribution.

**Baselines.** FedAvg McMahan et al. (2017) is proposed as the basic framework in the federated learning. And FedOpt improves it as a two-stage optimizer with a local and global optimizer update alternatively. Yang et al. (2021) proves a specific $\eta_g$ (not the average weight) can achieve faster convergence (non-dominant term). FedAdam Reddi et al. (2020) utilizes a adaptive optimizer on the global server and SGD optimizer on the clients, which average the averaged local gradients as a quasi-gradient for global server to implement the adaptive update. SCAFFOLD Karimireddy et al. (2020) applies the variance reduction technique , i.e. SVRG, to approximate the global gradient as the averaged local gradients and transfer an extra variable to the client per round. This implementation can accelerate the convergence rate of the non-dominant term theoretically and achieve a high performance empirically. FedCM Xu et al. (2021) proposes a client-level momentum to merge the global update as a momentum buffer to the local updates, which extremely reduces the local consistency. Though it introduces a unpredictable biases into the local updates, it achieves the SOTA performance ahead of other methods. FedProx Sahu et al. (2018) implements the prox-point optimizer into the FL framework on local updates with a regularization prox-term regularizer. It limits the local updates towards the initial point at the start of each local stage. Many previous works have analyzed its advantages and weaknesses. Durmus et al. (2021); Wang et al. (2022); Gong et al. (2022) use different variants of primal-dual method into FL and achieve nice satisfactory in the FL framework. It does not need a heterogeneity bounded assumption theoretically, which requires a high local convergence guarantees. Our proposed FedSpeed achieve the same convergence rate without assuming the local exact solution and we provide the local interval bound to achieve this faster convergence. Both theoretical analysis and empirical results verifies the performance of our proposed FedSpeed.

## B.2 EXPERIMENTS

## B.3 HYER-PARAMETERS

**Hyper-parameters Selections.** We fix the local learning rate as 0.1 and global learning rate as 1.0 for average, except for the FedAdam which is applied 0.1. The penalized weight of prox-term in FedProx, FedDyn, FedADMM and FedSpeed is selected from the [0.001, 0.01, 0.1, 0.5]. The learning rate decay is fixed as 0.998 expect for the FedDyn, FedADMM and FedSpeed is selected from [0.998, 0.999, 0.9995, 0.99995]. The perturbation weight is selected from [0, 0.5, 0.75, 0.875, 0.9375, 1]. The batchsize is selected from [20, 50]. The local interval $K$ is selected from [1, 2, 5, 10, 20]. For the specific parameters in FedAdam, the momentum weight is set as 0.1 and the second order momentum weight is set as 0.01. The minimal value is set as 0.001 to prevent the calculation of dividing by 0. The client-level momentum weight of FedCM is set as 0.1.

Table 4: Communication rounds required to achieve the target accuracy. On CIFAR-10/100 it trains 1,500 rounds and on TinyImagenet it trains 3,000 rounds. "-" means the test accuracy can not achieve the target accuracy within the fixed training rounds. **DIR** represents for the Dirichlet distribution with the heterogeneity weight equal to 0.6. Local interval $K$ is set as 5 on CIFAR-10 (100-10%) and 2 on others. Other hyper-parameters are introduced above.

| Dataset | CIFAR-10 (100-10%) | | | | CIFAR-10 (500-2%) | | | |
|---|---|---|---|---|---|---|---|---|
| Heterogeneity | IID. | | DIR. | | IID. | | DIR. | |
| Target Acc. (%) | 80.0 | 85.0 | 80.0 | 85.0 | 75.0 | 82.5 | 75.0 | 82.5 |
| FedAvg | 344 | - | 472 | - | 772 | - | 1357 | - |
| FedProx | 338 | - | 465 | - | 720 | - | 1151 | - |
| FedAdam | 324 | 1343 | 689 | - | 613 | 1476 | 878 | - |
| SCAFFOLD | 207 | 654 | 272 | - | 628 | - | 967 | - |
| FedCM | **109** | 620 | 192 | 1092 | **325** | 1160 | **449** | 1399 |
| FedDyn | **121** | **400** | 166 | - | 547 | - | 673 | - |
| FedADMM | 169 | 917 | **174** | **756** | 505 | 1440 | 687 | - |
| **FedSpeed** | 136 | **280** | 169 | 380 | 495 | 926 | 662 | 1148 |

| Dataset | CIFAR-100 (500-2%) | | | | TinyImagenet (500-2%) | | | |
|---|---|---|---|---|---|---|---|---|
| Heterogeneity | IID. | | DIR. | | IID. | | DIR. | |
| Target Acc. | 40.0 | 50.0 | 40.0 | 50.0 | 33.0 | 40.0 | 33.0 | 40.0 |
| FedAvg | 1013 | - | - | - | 1615 | - | - | - |
| FedProx | 957 | - | - | - | 1588 | - | - | - |
| FedAdam | 614 | 1277 | 847 | - | 1151 | 2495 | 1584 | - |
| SCAFFOLD | 720 | - | 784 | - | 949 | - | 1187 | - |
| FedCM | **505** | 1150 | 526 | 1336 | 661 | 1360 | 817 | 1843 |
| FedDyn | 661 | - | 703 | - | 1419 | - | 2559 | - |
| FedADMM | 687 | - | 715 | - | 921 | - | 2711 | - |
| **FedSpeed** | 522 | **973** | **541** | **1038** | **684** | **1373** | **962** | **1885** |

Here we briefly introduce the selection of the hyperparameters in FedSpeed.

(1) $\eta_l$ is the learning rate which is a basic hyperparameters in the deep learning, and usually we do not finetune this for the fair comparison in the experiments. We just select the same and common settings as the previous works mentioned.

(2) $\lambda$ is the coefficient for the prox-term, which is proposed in the FedProx and a lot of prox-based federated methods adopt this hyperparameter widely both in personalized-FL and centralized-FL. The selection of this hyperparameter has been studied in many previous works which verify its efficiency. Usually the selection of $\lambda$ are in $\{10, 100\}$ on the CIFAR-10/100 dataset, and we test it also works on the TinyImagenet.

(3) $\rho$ is the ascent step learning rate. Like many extra gradient method, the selection of $\rho$ is usually related to the local learning rate $\eta_l$. In order not to unduly affect the performance of the gradient descent, the learning rate for the extra gradient step $\rho$ is usually set not much larger than the learning rate for the gradient descent step $\eta_l$. Obviously, if $\rho$ is set very small, the updated state of the extra gradient steps will be very limited, which makes this operation have no effect. Therefore, the selection

of $\rho$ usually matches that of $\eta_l$. In our experiments, the $\eta_l$ is set as $0.1$, which is a common selection in the previous works. We test the selection of $\rho$ in $\{0, 0.01, 0.05, 0.1, 0.2\}$ which represents for $\{$"no extra gradient", "$0.1\eta_l$", "$0.5\eta_l$", "$1\eta_l$", "$2\eta_l$"$\}$. The best performing selection is $\rho = 1\eta_l$ in CIFAR-10 (details in Section 5.3 paragraph "Learning rate $\rho$ for the gradient perturbation"). We also test this selection on the CIFAR-100 and TinyImagenet, and it also works well. We recommend that the selection of $\rho$ should be kept comparable to the learning rate $\eta_l$.

(4) $\alpha$ is the ratio for merging the gradient of the extra ascent step. In FedSpeed, the $\alpha$ is in the range of $[0, 1]$. The same, if $\alpha$ is set very small, which means it does not merge the gradient of the ascent steps. In our experiments, we test the selection of $\alpha$ in $\{0, 0.5, 0.75, 0.875, 0.9375, 1.0\}$. The best performing selection is $\alpha = 0.9375$ in CIFAR-10. In fact $\alpha = 1$ also works well (details in Section 5.3 paragraph "Perturbation weight $\alpha$"). Thus, about $\alpha$, we recommend that it should be close to $1.0$, e.g. for $0.9, 0.99, 1.0$. This also verifies the improvements of the ascent steps.

### B.3.1 BEST PERFORMING HYPER-PARAMETERS.

For fair comparison, the learning rate is fixed for all the methods.

For CIFAR-10 dataset, we select the batchsize as 50 for 100 clients and 20 for 500 clients. The total dataset is 50,000 and there are 100 images under a single client if it is set as 500 clients. Thus we decay it to 20 for 5 iterations per local epoch. The local epochs is set as 5, the same as the experiments of Karimireddy et al. (2020); Durmus et al. (2021); Xu et al. (2021) etc. and their performance is matching. We select the local interval $K$ as 5. The prox-term weight is selected as 0.1. The learning rate decay is selected as 0.9995 for prox-term based methods. We train the total dataset for 1,500 communication rounds.

For CIFAR-100 dataset, we select the 500 clients with $2\%$ participation ratio in the experiments. Thus for each hyper-parameters we fine-tune a little. The batchsize is selected as 20 to avoid too little iterations per local epoch. The local epochs is set as 2 for the final results comparison. The ablation study on local interval $K$ indicates that our proposed FedSpeed outperforms significantly than other methods when $K$ is large. Thus to compare the performance more clearly, we select the 2 as the local epochs. We decay the prox-term weight as 0.01 for prox-term based methods. The learning rate decay is selected as 0.99995 for prox-based methods. We train 1,500 rounds and then test the performance.

For TinyImagenet dataset, the most selections are the same as for the CIFAR-100 dataset. The prox-term weight is selected as 0.1 and the learning rate decay is selected as 0.9995. Total 3,000 communication rounds are implemented in the training stage.

### B.3.2 SPEED COMPARISON.

Table 4 shows the communication rounds required to achieve the target test accuracy. At the beginning of training, FedCM performs faster than others and usually achieve a high accuracy finally. FedSpeed is faster in the middle and late stages of training. We bold the data for the top-2 in each test and generally FedCM and FedSpeed significantly performs well on the training speed.

Figure 4 shows the performance of different learning rate decay and prox-term weight for FedSpeed.

### B.3.3 TIME COST

Table 5: Training wall-clock time comparison.

| $\alpha_1$ | Times (s/Round) | Rounds | Total (s) | Cost Ratio |
|------------|-----------------|--------|-----------|------------|
| FedAvg     | 10.44           | -      | -         | -          |
| FedProx    | 11.33           | -      | -         | -          |
| FedAdam    | 14.74           | 1343   | 19795.8   | 4.31×      |
| SCAFFOLD   | 14.34           | 654    | 9378.3    | 2.03×      |
| FedCM      | 13.22           | 622    | 8222.8    | 1.78×      |
| FedDyn     | 14.11           | 400    | 5644.0    | 1.22×      |
| FedSpeed   | 16.42           | 281    | 4614.0    | 1×         |

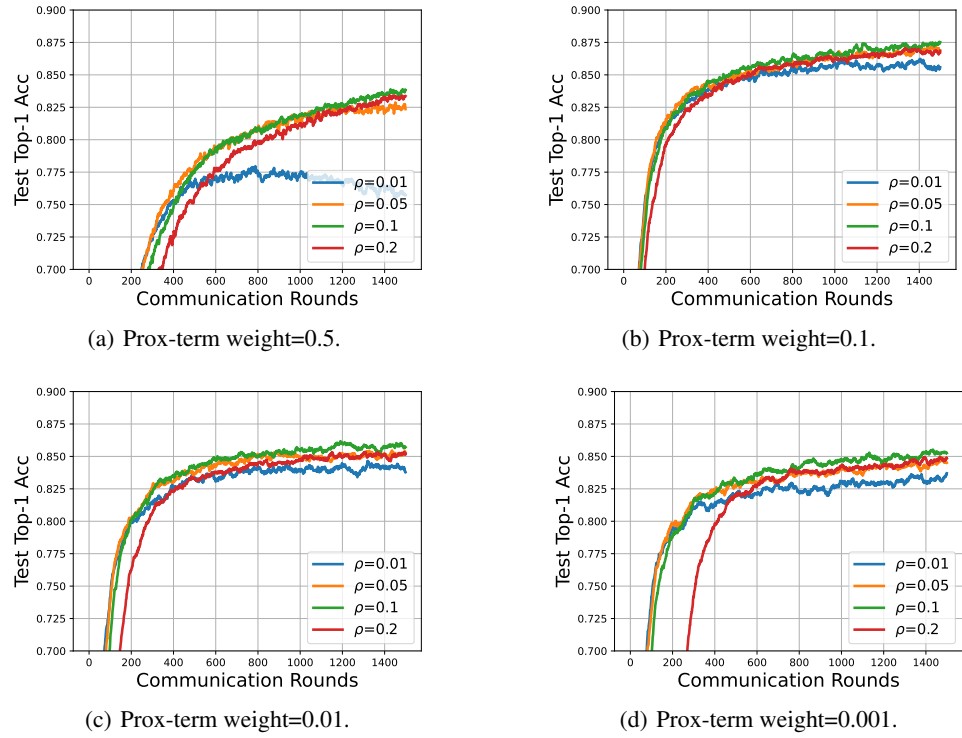

Figure 4: Performance of different ascent step size $\rho$ under different prox-term weights of $[0.001, 0.01, 0.1, 0.5]$.

We test the time on the A100-SXM4-40GB GPU and show the performance in the Table B.3.3. Experimental setups are the same as the CIFAR-10 $10\%$ participation among total 100 clients on the DIR-0.6 dataset. The rounds in the table are the communication rounds required that the test accuracy achieves accuracy $85\%$. "-" means it can not achieve the target accuracy.

FedSpeed is slower due to the requirement of computing an extra gradient. So it gets slower in one single update, approximately $1.57\times$ wall-clock time costs than FedAvg. But its convergence process is very fast. For the final convergence speed, FedSpeed still has a considerable advantage over other algorithms. The issue is possibly one of the improvements for FedSpeed in the future. For example, introduces a single-call gradient method to save half the costs during backpropagation. We are also currently trying to introduce new module to save the cost.

## B.3.4 DIFFERENT HETEROGENEITY.

Table 6: Comparison on different heterogeneous dataset.

| $\alpha_1$ | IID | Dir-0.6 | Dir-0.3 | Drops (i.i.d. $>$ Dir-0.6) | Drops (Dir-0.6 $>$ Dir-0.3) |
|---|---|---|---|---|---|
| FedAvg | 77.01 | 75.21 | 71.96 | 1.80 | 3.25 |
| FedAdam | 82.92 | 80.55 | 76.87 | 2.37 | 3.68 |
| SCAFFOLD | 80.11 | 77.71 | 74.34 | 2.40 | 3.37 |
| FedCM | 84.20 | 83.48 | 81.02 | 0.72 | 2.46 |
| FedDyn | 83.36 | 80.57 | 77.33 | 2.79 | 3.24 |
| FedSpeed | 85.80 | 84.79 | 82.68 | 1.01 | 2.11 |

We test on the Dir-0.3 setups on CIFAR-10 and show the results as Table B.3.4, the other settings are the same as the test in the text. The (i.i.d. $>$ Dir-0.6) is the difference between the IID dataset and the Dir-0.6 dataset and (Dir-0.6 $>$ Dir-0.3) is the difference between the Dir-0.6 dataset and the DIR-0.3 dataset. FedSpeed can outperform the others on the Dir-0.3 setups whose heterogeneity is much

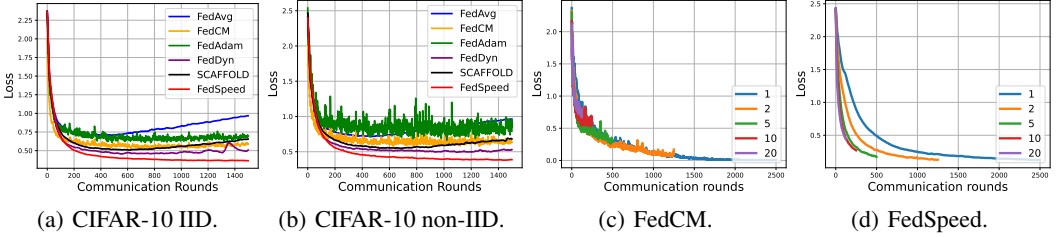

| (a) CIFAR-10 IID. | (b) CIFAR-10 non-IID. | (c) FedCM. | (d) FedSpeed. |

Figure 5: (a) and (b) show the loss curves on the CIFAR-10 IID/DIR-0.6 dataset. The FedSpeed achieves the best and stable performance in the training. (c) and (d) show the loss curve of FedCM and FedSpeed on the CIFAR-10 DIR-0.6 dataset with increasing the local epochs from $E = 1$ to 20.

stronger than Dir-0.6 setups. the heterogeneity becomes stronger, FedSpeed can still maintain a stable generalization performance. The correction term helps to correct the biases during the local training, while the gradient perturbation term helps to resist the local over-fitting on the heterogeneous dataset. FedSpeed can benefit from avoiding falling into the biased optima.

### B.3.5 ABLATION STUDIES

Table 7: Ablation studies on different modules.

| Prox-term | Prox-correction term | Gradient perturbation | Accuracy (%) |
|-----------|----------------------|-----------------------|--------------|
| - | - | - | 81.92 |
| $\checkmark$ | - | - | 82.24 |
| $\checkmark$ | $\checkmark$ | - | 83.94 |
| $\checkmark$ | - | $\checkmark$ | 83.88 |
| $\checkmark$ | $\checkmark$ | $\checkmark$ | 85.70 |

From the practical training point of view, compared with the vanilla FedAvg, FedSpeed adds three main modules: (1) prox-term, (2) prox-correction term, and (3) gradient perturbation. We test the performance of 500 communication rounds of the different combination of the modules above on the CIFAR-10 with the settings of 10% participating ratio of total 100 clients. The TableB.3.5 shows their performance.

From the table above, we can clearly see the performance of different modules. The prox-term is proposed by the FedProx. But due to some issues we point out in our paper, this term has also a negative impact on the performance in FL. When the prox-correction term is introduced in, it improves the performance from 82.24% to 83.94%. When the gradient perturbation is introduced in, it improves the performance from 82.24% to 83.88%. While FedSpeed applies them together and achieves a 3.46% improvement.

Different performance of these modules:

As introduced in our paper, the prox-term simply performs as a balance between the local and global solutions, and there still exists the non-vanishing inconsistent biases among the local solutions, i.e., the local solutions are still largely deviated from each other, implying that local inconsistency is still not eliminated. Thus we utilize the prox-correction term to correct the inconsistent biases during the local training. About the function of gradient perturbation, we refer to a theoretical explanation in the main text, and its proof is provided in the supplementary material due to the space limitations. This perturbation is similar to utilize a penalized gradient term to the objective function during local optimization process. The additional penalty will bring better properties to the local state, e.g. for flattened minimal and smoothness. For federated learning, the smoother the local minima is, the more flatness the model merged on the server will be. FedSpeed benefits from these two modules to improve the performance and achieves the SOTA results.

### B.3.6 Loss curve comparison

According to the Figure 5, in (a) and (b) we can see the common FedAvg method fail to resist the local over-fitting and finally does not approach a stable state in the training, while FedSpeed can converge stably and efficiently, which if far faster than other baselines. (c) and (d) show the empirical studies of increasing the local interval. FedCM is the second SOTA-performing method in our baselines, while it still can not speed up by increasing local interval in the practical training. As shown in (c), increasing local interval almost does not have any benefits to FedCM, and the communication rounds can not be reduced. While in (d), FedSpeed succeed to apply a larger local interval to reduce the communication rounds. When $K$ is increasing, to achieve the similar performance, the total communication rounds is nearly decreasing as $K\times$, which is a useful property that is very efficient in practical training.

In our paper, what we claim is that if a federated learning method could adopt a large $K$, it is a good property. Unfortunately, most SGD-type algorithms cannot increase the convergence rate by increasing $K$. Some useful techniques are adopt in the FL framework to improve the performance, e.g. variance reduction, gradient tracking and regularization term (mainly the prox-based methods). FedSpeed is a prox-based method which incorporates the correction term and extra ascent gradient to improve the performance. In fact, it has been proven that prox-based methods have the potential to apply the larger local interval in the local training under the requirement of local minimal solution per communication round. We theoretically prove that FedSpeed can achieve the fast rate without this harsh assumption and it can apply the large $K$ in the local client. Yang et al. (2021) have proven that if FedAvg change the partial participation to the full participation (Local-SGD-type), the dominant term of convergence rate will change from $\mathcal{O}(\frac{\sqrt{K}}{\sqrt{nT}})$ to $\mathcal{O}(\frac{1}{\sqrt{mKT}})$, which will be relaxed to $K$ times faster. Full participation usually achieves higher theoretical rate than the partial participation. VRL-SGD Liang et al. (2019) can theoretically improve the efficiency by adopting a larger order of local interval $K = \mathcal{O}(\sqrt{T})$ than FedAvg, while FedSpeed can adopt $K = \mathcal{O}(T)$.

### B.3.7 Ablation Study of Perturbation weight $\alpha$

**Perturbation weight $\alpha$.** $\alpha$ determines the degree of influence of the perturbation gradient term to the vanilla stochastic gradient on the local training stage. It is a trade-off to balance the ratio of the perturbation term. We select the $\alpha$ from 0 to 1 and find FedSpeed can converge with any $\alpha \in [0, 1]$. Though the theoretical analysis demonstrates that by applying a $\alpha > 0$ in the term $\Phi$ will not increasing the extra orders. And the experimental results shown in Table 8, indicates that the generalization performance improves by increasing $\alpha$.

Table 8: Performance of different $\alpha$ with $\rho_0 = 0.1$.

| $\alpha$ | 0 | 0.5 | 0.75 | 0.875 | **0.9375** | 1.0 |
|---|---|---|---|---|---|---|
| Acc. | 83.97 | 84.36 | 84.91 | 85.46 | **85.74** | 85.72 |

## C Proofs for Analysis

In this part we will demonstrate the proofs of all formula mentioned in this paper. Each formula is presented in the form of a lemma.

### C.1 Proof of Equation (2)

Equation (2) shows the update in the total local training stage.

**Lemma C.1** *For $\forall \, \mathbf{x}_{i,k}^t \in \mathbb{R}^d$ and $i \in \mathcal{S}^t$, we denote $\delta_{i,k}^t = \mathbf{x}_{i,k}^t - \mathbf{x}_{i,k-1}^t$ with setting $\delta_{i,0}^t = 0$, and $\Delta_{i,K}^t = \sum_{k=0}^{K} \delta_{i,k}^t = \mathbf{x}_{i,K}^t - \mathbf{x}_{i,0}^t$, under the update rule in Algorithm Algorithm 1, we have:*

$$\Delta_{i,K}^t = -\lambda\gamma \sum_{k=0}^{K-1} \frac{\gamma_k}{\gamma} \tilde{\mathbf{g}}_{i,k}^t + \gamma\lambda\hat{\mathbf{g}}_i^{t-1}, \tag{11}$$

*where $\sum_{k=0}^{K-1} \gamma_k = \sum_{k=0}^{K-1} \frac{\eta_l}{\lambda}\left(1 - \frac{\eta_l}{\lambda}\right)^{K-1-k} = \gamma = 1 - \left(1 - \frac{\eta_l}{\lambda}\right)^K$.*

**Proof 1** *According to the update rule of Line.11 in Algorithm Algorithm 1, we have:*

$$\delta_k = \Delta_{i,k}^t - \Delta_{i,k-1}^t = \mathbf{x}_{i,k}^t - \mathbf{x}_{i,k-1}^t$$
$$= -\eta_l\big(\tilde{\mathbf{g}}_{i,k-1}^t - \hat{\mathbf{g}}_i^{t-1} + \frac{1}{\lambda}(\mathbf{x}_{i,k-1}^t - \mathbf{x}_{i,0}^t)\big) = -\eta_l(\tilde{\mathbf{g}}_{i,k-1}^t - \hat{\mathbf{g}}_i^{t-1} + \frac{1}{\lambda}\Delta_{i,k-1}^t).$$

*Then We can formulate the iterative relationship of $\Delta_{i,k}^t$ as:*

$$\Delta_{i,k}^t = \Delta_{i,k-1}^t - \eta_l(\tilde{\mathbf{g}}_{i,k-1}^t - \hat{\mathbf{g}}_i^{t-1} + \frac{1}{\lambda}\Delta_{i,k-1}^t) = (1 - \frac{\eta_l}{\lambda})\Delta_{i,k-1}^t - \eta_l(\tilde{\mathbf{g}}_{i,k-1}^t - \hat{\mathbf{g}}_i^{t-1}).$$

*Taking the iteration on $k$ and we have:*

$$\mathbf{x}_{i,K}^t - \mathbf{x}_{i,0}^t = \Delta_{i,K}^t = (1 - \frac{\eta_l}{\lambda})^K \Delta_{i,0}^t - \eta_l \sum_{k=0}^{K-1}(1 - \frac{\eta_l}{\lambda})^{K-1-k}(\tilde{\mathbf{g}}_{i,k}^t - \hat{\mathbf{g}}_i^{t-1})$$

$$\overset{(a)}{=} -\eta_l \sum_{k=0}^{K-1}(1 - \frac{\eta_l}{\lambda})^{K-1-k}(\tilde{\mathbf{g}}_{i,k}^t - \hat{\mathbf{g}}_i^{t-1})$$

$$= -\lambda \sum_{k=0}^{K-1} \frac{\eta_l}{\lambda}(1 - \frac{\eta_l}{\lambda})^{K-1-k}(\tilde{\mathbf{g}}_{i,k}^t - \hat{\mathbf{g}}_i^{t-1})$$

$$= -\lambda \sum_{k=0}^{K-1} \frac{\eta_l}{\lambda}(1 - \frac{\eta_l}{\lambda})^{K-1-k}\tilde{\mathbf{g}}_{i,k}^t + \big(1 - (1 - \frac{\eta_l}{\lambda})^K\big)\lambda\hat{\mathbf{g}}_i^{t-1}$$

$$= -\lambda\gamma \sum_{k=0}^{K-1} \frac{\gamma_k}{\gamma}\tilde{\mathbf{g}}_{i,k}^t + \gamma\lambda\hat{\mathbf{g}}_i^{t-1}.$$

*(a) applies $\Delta_{i,0}^t = \delta_{i,0}^t = 0$.*

## C.2 PROOF OF EQUATION (3)

Equation (3) shows the update of the prox-correction term, which utilizes the weighted sum of the previous local offsets as a bias controller for eliminating the non-vanishing bias resulting from the prox-term.

**Lemma C.2** *Under the update rule in Algorithm Algorithm 1, we have:*

$$\hat{\mathbf{g}}_i^t = (1 - \gamma)\hat{\mathbf{g}}_i^{t-1} + \gamma \sum_{k=0}^{K-1} \frac{\gamma_k}{\gamma}\tilde{\mathbf{g}}_{i,k}^t. \tag{12}$$

*where $\sum_{k=0}^{K-1}\gamma_k = \sum_{k=0}^{K-1} \frac{\eta_l}{\lambda}\big(1 - \frac{\eta_l}{\lambda}\big)^{K-1-k} = \gamma = 1 - (1 - \frac{\eta_l}{\lambda})^K$.*

**Proof 2** *According to the update rule of Line.13 in Algorithm Algorithm 1, we have:*

$$\hat{\mathbf{g}}_i^t = \hat{\mathbf{g}}_i^{t-1} - \frac{1}{\lambda}(\mathbf{x}_{i,K}^t - \mathbf{x}_{i,0}^t)$$

$$\overset{(a)}{=} \hat{\mathbf{g}}_i^{t-1} + \frac{\eta_l}{\lambda} \sum_{k=0}^{K-1}\big(1 - \frac{\eta_l}{\lambda}\big)^{K-1-k}(\tilde{\mathbf{g}}_{i,k}^t - \hat{\mathbf{g}}_i^{t-1})$$

$$= \hat{\mathbf{g}}_i^{t-1} + \frac{\eta_l}{\lambda} \sum_{k=0}^{K-1}\big(1 - \frac{\eta_l}{\lambda}\big)^{K-1-k}\tilde{\mathbf{g}}_{i,k}^t - \frac{\eta_l}{\lambda}\Big(\sum_{k=0}^{K-1}\big(1 - \frac{\eta_l}{\lambda}\big)^{K-1-k}\Big)\hat{\mathbf{g}}_i^{t-1}$$

$$= \hat{\mathbf{g}}_i^{t-1} + \frac{\eta_l}{\lambda} \sum_{k=0}^{K-1}\big(1 - \frac{\eta_l}{\lambda}\big)^{K-1-k}\tilde{\mathbf{g}}_{i,k}^t - \frac{\eta_l}{\lambda} \frac{1 - (1 - \frac{\eta_l}{\lambda})^K}{\frac{\eta_l}{\lambda}}\hat{\mathbf{g}}_i^{t-1}$$

$$= (1 - \frac{\eta_l}{\lambda})^K \hat{\mathbf{g}}_i^{t-1} + \frac{\eta_l}{\lambda} \sum_{k=0}^{K-1}\big(1 - \frac{\eta_l}{\lambda}\big)^{K-1-k}\tilde{\mathbf{g}}_{i,k}^t$$

$$= (1 - \gamma)\hat{\mathbf{g}}_i^{t-1} + \gamma \sum_{k=0}^{K-1} \frac{\gamma_k}{\gamma} \tilde{\mathbf{g}}_{i,k}^t.$$

*(a) applies the Lemma C.1.*

## C.3 Proof of Equation (4) and (5)

**Lemma C.3** *Considering the $\mathbf{u}^{t+1} = \frac{1}{m} \sum_{i \in [m]} \mathbf{x}_{i,K}^t$ is the mean averaged parameters among the last iteration of local clients at time $t$, the auxiliary sequence $\{\mathbf{z}^t = \mathbf{u}^t + \frac{1-\gamma}{\gamma}(\mathbf{u}^t - \mathbf{u}^{t-1})\}_{t>0}$ satisfies the update rule as:*

$$\mathbf{z}^{t+1} = \mathbf{z}^t - \lambda \frac{1}{m} \sum_{i \in [m]} \sum_{k=0}^{K-1} \frac{\gamma_k}{\gamma} \tilde{\mathbf{g}}_{i,k}^t. \tag{13}$$

**Proof 3** *Firstly, according to the lemma C.1 and Line.14 and Line.16 in Algorithm 1, we have:*

$$\mathbf{u}^{t+1} - \mathbf{u}^t = \frac{1}{m} \sum_{i \in [m]} (\mathbf{x}_{i,K}^t - \mathbf{x}_{i,K}^{t-1})$$

$$= \frac{1}{m} \sum_{i \in [m]} (\mathbf{x}_{i,K}^t - \mathbf{x}_{i,0}^t - \lambda \hat{\mathbf{g}}_i^{t-1})$$

$$= \frac{1}{m} \sum_{i \in [m]} (-\lambda \gamma \sum_{k=0}^{K-1} \frac{\gamma_k}{\gamma} \tilde{\mathbf{g}}_{i,k}^t + \lambda \gamma \hat{\mathbf{g}}_i^t - \lambda \hat{\mathbf{g}}_i^{t-1})$$

$$= -\lambda \frac{1}{m} \sum_{i \in [m]} \sum_{k=0}^{K-1} \frac{\gamma_k}{\gamma} (\gamma \tilde{\mathbf{g}}_{i,k}^t + (1 - \gamma)\hat{\mathbf{g}}_i^{t-1}).$$

*This could be considered as a momentum-like term with the coefficient of $\gamma$. Here we define a virtual observation sequence $\{\mathbf{u}^t\}$ and its update rule is:*

$$\mathbf{u}_{i,k+1}^t = \mathbf{u}_{i,k}^t - \lambda \frac{\gamma_k}{\gamma} (\gamma \tilde{\mathbf{g}}_{i,k}^t + (1 - \gamma)\hat{\mathbf{g}}_i^{t-1}),$$

$$\mathbf{u}_{i,0}^{t+1} = \mathbf{u}^{t+1} = \frac{1}{m} \sum_{i \in [m]} \mathbf{u}_{i,K}^t.$$

*According to the lemma C.2 and above update rule, we can get that:*

$$\hat{\mathbf{g}}_i^t = (1 - \gamma)\hat{\mathbf{g}}_i^{t-1} + \gamma \sum_{k=0}^{K-1} \frac{\gamma_k}{\gamma} \tilde{\mathbf{g}}_{i,k}^t$$

$$= -\frac{1}{\lambda}(\mathbf{u}_{i,K}^t - \mathbf{u}_{i,0}^t) - \gamma \sum_{k=0}^{K-1} \frac{\gamma_k}{\gamma} \tilde{\mathbf{g}}_{i,k}^t + \gamma \sum_{k=0}^{K-1} \frac{\gamma_k}{\gamma} \tilde{\mathbf{g}}_{i,k}^t = -\frac{1}{\lambda}(\mathbf{u}_{i,K}^t - \mathbf{u}_{i,0}^t).$$

*This function indicates that the virtual sequence $\mathbf{u}^t$ could be considered as a momentum-based update method with a global correction term to guide the local update, and the correction term is calculated from the offset of the virtual observation sequence during the training process at round $t$.*

*Then we expand the the auxiliary sequence $\mathbf{z}^t$ as:*

$$\mathbf{z}^{t+1} - \mathbf{z}^t = (\mathbf{u}^{t+1} - \mathbf{u}^t) + \frac{1 - \gamma}{\gamma}(\mathbf{u}^{t+1} - \mathbf{u}^t) - \frac{1 - \gamma}{\gamma}(\mathbf{u}^t - \mathbf{u}^{t-1})$$

$$
\begin{aligned}
&= \frac{1}{\gamma}(\mathbf{u}^{t+1} - \mathbf{u}^t) - \frac{1-\gamma}{\gamma}(\mathbf{u}^t - \mathbf{u}^{t-1}) \\
&= -\lambda\frac{1}{m}\sum_{i\in[m]}\left(\left(\sum_{k=0}^{K-1}\frac{\gamma_k}{\gamma}\tilde{\mathbf{g}}_{i,k}^t\right) + \frac{1-\gamma}{\gamma}\hat{\mathbf{g}}_i^{t-1}\right) - \frac{1-\gamma}{\gamma}(\mathbf{u}^t - \mathbf{u}^{t-1}) \\
&= -\lambda\frac{1}{m}\sum_{i\in[m]}\sum_{k=0}^{K-1}\frac{\gamma_k}{\gamma}\tilde{\mathbf{g}}_{i,k}^t - \frac{1-\gamma}{\gamma}\frac{1}{m}\sum_{i\in[m]}\lambda\hat{\mathbf{g}}_i^{t-1} - \frac{1-\gamma}{\gamma}(\mathbf{u}^t - \mathbf{u}^{t-1}) \\
&= -\lambda\frac{1}{m}\sum_{i\in[m]}\sum_{k=0}^{K-1}\frac{\gamma_k}{\gamma}\tilde{\mathbf{g}}_{i,k}^t - \frac{1-\gamma}{\gamma}\frac{1}{m}\sum_{i\in[m]}(\mathbf{u}^t - \mathbf{u}^{t-1} + \lambda\hat{\mathbf{g}}_i^{t-1}) \\
&= -\lambda\frac{1}{m}\sum_{i\in[m]}\sum_{k=0}^{K-1}\frac{\gamma_k}{\gamma}\tilde{\mathbf{g}}_{i,k}^t - \frac{1-\gamma}{\gamma}\frac{1}{m}\sum_{i\in[m]}(\mathbf{x}_{i,K}^{t-1} - \mathbf{x}_{i,K}^{t-2} + \lambda\hat{\mathbf{g}}_i^{t-1}) \\
&= -\lambda\frac{1}{m}\sum_{i\in[m]}\sum_{k=0}^{K-1}\frac{\gamma_k}{\gamma}\tilde{\mathbf{g}}_{i,k}^t - \frac{1-\gamma}{\gamma}\frac{1}{m}\sum_{i\in[m]}(\mathbf{x}_{i,K}^{t-1} - \mathbf{x}_{i,0}^{t-1} + \lambda\hat{\mathbf{g}}_i^{t-1} - \lambda\hat{\mathbf{g}}_i^{t-2}) \\
&= -\lambda\frac{1}{m}\sum_{i\in[m]}\sum_{k=0}^{K-1}\frac{\gamma_k}{\gamma}\tilde{\mathbf{g}}_{i,k}^t.
\end{aligned}
$$

## C.4 PROOF OF THEOREM 4.5

Firstly we state some important lemmas applied in the proof.

**Lemma C.4** *(Bounded global update) The global update $\frac{1}{m}\sum_{i\in[m]}\hat{\mathbf{g}}_i^t$ holds the upper bound of:*

$$
\mathbb{E}_t\|\frac{1}{m}\sum_{i\in[m]}\hat{\mathbf{g}}_i^{t-1}\|^2 \leq \frac{1}{\gamma}\left(\mathbb{E}_t\|\frac{1}{m}\sum_{i\in[m]}\hat{\mathbf{g}}_i^{t-1}\|^2 - \mathbb{E}_t\|\frac{1}{m}\sum_{i\in[m]}\hat{\mathbf{g}}_i^t\|^2\right) + \mathbb{E}_t\|\frac{1}{m}\sum_{i\in[m]}\sum_{k=0}^{K-1}\frac{\gamma_k}{\gamma}\tilde{\mathbf{g}}_{i,k}^t\|^2.
$$

**Proof 4** *According to the lemma C.2,we have:*

$$
\frac{1}{m}\sum_{i\in[m]}\hat{\mathbf{g}}_i^t = (1-\gamma)\frac{1}{m}\sum_{i\in[m]}\hat{\mathbf{g}}_i^{t-1} + \gamma\frac{1}{m}\sum_{i\in[m]}\sum_{k=0}^{K-1}\frac{\gamma_k}{\gamma}\tilde{\mathbf{g}}_{i,k}^t.
$$

*Take the L2-norm and we have:*

$$
\begin{aligned}
\|\frac{1}{m}\sum_{i\in[m]}\hat{\mathbf{g}}_i^t\|^2 &= \|(1-\gamma)\frac{1}{m}\sum_{i\in[m]}\hat{\mathbf{g}}_i^{t-1} + \gamma\frac{1}{m}\sum_{i\in[m]}\sum_{k=0}^{K-1}\frac{\gamma_k}{\gamma}\tilde{\mathbf{g}}_{i,k}^t\|^2 \\
&\leq (1-\gamma)\|\frac{1}{m}\sum_{i\in[m]}\hat{\mathbf{g}}_i^{t-1}\|^2 + \gamma\|\frac{1}{m}\sum_{i\in[m]}\sum_{k=0}^{K-1}\frac{\gamma_k}{\gamma}\tilde{\mathbf{g}}_{i,k}^t\|^2.
\end{aligned}
$$

*Thus we have the following recursion,*

$$
\mathbb{E}_t\|\frac{1}{m}\sum_{i\in[m]}\hat{\mathbf{g}}_i^{t-1}\|^2 \leq \frac{1}{\gamma}\left(\mathbb{E}_t\|\frac{1}{m}\sum_{i\in[m]}\hat{\mathbf{g}}_i^{t-1}\|^2 - \mathbb{E}_t\|\frac{1}{m}\sum_{i\in[m]}\hat{\mathbf{g}}_i^t\|^2\right) + \mathbb{E}_t\|\frac{1}{m}\sum_{i\in[m]}\sum_{k=0}^{K-1}\frac{\gamma_k}{\gamma}\tilde{\mathbf{g}}_{i,k}^t\|^2.
$$

**Lemma C.5** *(Bounded local update) The local update $\hat{\mathbf{g}}_i^t$ holds the upper bound of:*

$$\frac{1}{m}\sum_{i\in[m]}\mathbb{E}_t\|\hat{\mathbf{g}}_i^{t-1}\|^2 \leq \frac{P}{\gamma}\frac{1}{m}\sum_{i\in[m]}\left(\mathbb{E}_t\|\hat{\mathbf{g}}_i^{t-1}\|^2 - \mathbb{E}_t\|\hat{\mathbf{g}}_i^t\|^2\right) + \frac{24PL^2}{m}\sum_{i\in[m]}\sum_{k=0}^{K-1}\frac{\gamma_k}{\gamma}\mathbb{E}_t\|\mathbf{x}_{i,k}^t - \mathbf{x}^t\|^2$$
$$+ 12P\mathbb{E}_t\|\nabla F(\mathbf{z}^t)\|^2 + P(12\sigma_g^2 + \sigma_l^2),$$

*where $\frac{1}{P} = 1 - \frac{24\lambda^2 L^2 (1-2\gamma)^2}{\gamma^2}$.*

**Proof 5** *According to the lemmaC.2, we have:*

$$\hat{\mathbf{g}}_i^t = (1-\gamma)\hat{\mathbf{g}}_i^{t-1} + \gamma\sum_{k=0}^{K-1}\frac{\gamma_k}{\gamma}\tilde{\mathbf{g}}_{i,k}^t.$$

*Take the L2-norm and we have:*

$$\|\hat{\mathbf{g}}_i^t\|^2 = \|(1-\gamma)\hat{\mathbf{g}}_i^{t-1} + \gamma\sum_{k=0}^{K-1}\frac{\gamma_k}{\gamma}\tilde{\mathbf{g}}_{i,k}^t\|^2$$
$$\overset{(a)}{\leq} (1-\gamma)\|\hat{\mathbf{g}}_i^{t-1}\|^2 + \gamma\|\sum_{k=0}^{K-1}\frac{\gamma_k}{\gamma}\tilde{\mathbf{g}}_{i,k}^t\|^2$$
$$\overset{(b)}{\leq} (1-\gamma)\|\hat{\mathbf{g}}_i^{t-1}\|^2 + \gamma\sum_{k=0}^{K-1}\frac{\gamma_k}{\gamma}\|\tilde{\mathbf{g}}_{i,k}^t\|^2$$
$$= (1-\gamma)\|\hat{\mathbf{g}}_i^{t-1}\|^2 + \sum_{k=0}^{K-1}\gamma_k\|\tilde{\mathbf{g}}_{i,k}^t\|^2.$$

*(a) and (b) apply the Jensen inequality.*
*Thus we have the following recursion:*

$$\frac{1}{m}\sum_{i\in[m]}\mathbb{E}_t\|\hat{\mathbf{g}}_i^{t-1}\|^2 \leq \frac{1}{\gamma}\frac{1}{m}\sum_{i\in[m]}\left(\mathbb{E}_t\|\hat{\mathbf{g}}_i^{t-1}\|^2 - \mathbb{E}_t\|\hat{\mathbf{g}}_i^t\|^2\right) + \frac{1}{m}\sum_{i\in[m]}\sum_{k=0}^{K-1}\frac{\gamma_k}{\gamma}\mathbb{E}_t\|\tilde{\mathbf{g}}_{i,k}^t\|^2.$$

*Here we provide a loose upper bound as a constant for the quasi-stochastic gradient:*

$$\frac{1}{m}\sum_{i\in[m]}\sum_{k=0}^{K-1}\frac{\gamma_k}{\gamma}\mathbb{E}_t\|\tilde{\mathbf{g}}_{i,k}^t\|^2$$
$$= \frac{1}{m}\sum_{i\in[m]}\sum_{k=0}^{K-1}\frac{\gamma_k}{\gamma}\mathbb{E}_t\|(1-\alpha)\mathbf{g}_{i,k,1}^t + \alpha\mathbf{g}_{i,k,2}^t\|^2$$
$$= \frac{1}{m}\sum_{i\in[m]}\sum_{k=0}^{K-1}\frac{\gamma_k}{\gamma}\mathbb{E}_t\|\mathbf{g}_{i,k,1}^t + \alpha(\mathbf{g}_{i,k,2}^t - \mathbf{g}_{i,k,1}^t)\|^2$$
$$\leq \frac{2}{m}\sum_{i\in[m]}\sum_{k=0}^{K-1}\frac{\gamma_k}{\gamma}\left(\mathbb{E}_t\|\nabla F_i(\mathbf{x}_{i,k}^t)\|^2 + \alpha^2\mathbb{E}_t\|\nabla F_i(\breve{\mathbf{x}}_{i,k}^t) - \nabla F_i(\mathbf{x}_{i,k}^t)\|^2\right) + \sigma_l^2$$
$$\leq \frac{2}{m}\sum_{i\in[m]}\sum_{k=0}^{K-1}\frac{\gamma_k}{\gamma}\left(\mathbb{E}_t\|\nabla F_i(\mathbf{x}_{i,k}^t)\|^2 + \alpha^2 L^2\rho^2\mathbb{E}_t\|\nabla F_i(\mathbf{x}_{i,k}^t)\|^2\right) + \sigma_l^2$$
$$\leq \frac{4}{m}\sum_{i\in[m]}\sum_{k=0}^{K-1}\frac{\gamma_k}{\gamma}\mathbb{E}_t\|\nabla F_i(\mathbf{x}_{i,k}^t) - \nabla F_i(\mathbf{z}^t) + \nabla F_i(\mathbf{z}^t) - \nabla F(\mathbf{z}^t) + \nabla F(\mathbf{z}^t)\|^2 + \sigma_l^2$$

$$\leq \frac{12L^2}{m} \sum_{i \in [m]} \sum_{k=0}^{K-1} \frac{\gamma_k}{\gamma} \mathbb{E}_t \|\mathbf{x}_{i,k}^t - \mathbf{z}^t\|^2 + 12\mathbb{E}_t\|\nabla F(\mathbf{z}^t)\|^2 + (12\sigma_g^2 + \sigma_l^2)$$

$$\leq \frac{12L^2}{m} \sum_{i \in [m]} \sum_{k=0}^{K-1} \frac{\gamma_k}{\gamma} \mathbb{E}_t \|\mathbf{x}_{i,k}^t - \mathbf{x}^t + \mathbf{x}^t - \mathbf{u}^t + \mathbf{u}^t - \mathbf{z}^t\|^2$$
$$+ 12\mathbb{E}_t\|\nabla F(\mathbf{z}^t)\|^2 + (12\sigma_g^2 + \sigma_l^2)$$

$$\leq \frac{24L^2}{m} \sum_{i \in [m]} \sum_{k=0}^{K-1} \frac{\gamma_k}{\gamma} \mathbb{E}_t \|\mathbf{x}_{i,k}^t - \mathbf{x}^t\|^2 + 24L^2\|\mathbf{x}^t - \mathbf{u}^t + \mathbf{u}^t - \mathbf{z}^t\|^2 + (12\sigma_g^2 + \sigma_l^2)$$
$$+ 12\mathbb{E}_t\|\nabla F(\mathbf{z}^t)\|^2$$

$$\leq \frac{24L^2}{m} \sum_{i \in [m]} \sum_{k=0}^{K-1} \frac{\gamma_k}{\gamma} \mathbb{E}_t \|\mathbf{x}_{i,k}^t - \mathbf{x}^t\|^2 + \frac{24L^2\lambda^2(1-2\gamma)^2}{\gamma^2} \frac{1}{m} \sum_i \mathbb{E}_t\|\hat{\mathbf{g}}_i^{t-1}\|^2$$
$$+ 12\mathbb{E}_t\|\nabla F(\mathbf{z}^t)\|^2 + (12\sigma_g^2 + \sigma_l^2).$$

*We applies the Jensen inequality, the basic inequality $\|\sum_{i=1}^n \mathbf{a}_i\|^2 \leq n \sum_{i=1}^n \|\mathbf{a}_i\|^2$, and the upper bound of $\rho \leq \frac{1}{\alpha L}$. Combining the above inequalities, let $\frac{1}{P} = 1 - \frac{24L^2\lambda^2(1-2\gamma^2)}{\gamma^2}$ is the constant, we have:*

$$\frac{1}{m} \sum_{i \in [m]} \mathbb{E}_t\|\hat{\mathbf{g}}_i^{t-1}\|^2 \leq \frac{P}{\gamma} \frac{1}{m} \sum_{i \in [m]} \left( \mathbb{E}_t\|\hat{\mathbf{g}}_i^{t-1}\|^2 - \mathbb{E}_t\|\hat{\mathbf{g}}_i^t\|^2 \right) + \frac{24PL^2}{m} \sum_{i \in [m]} \sum_{k=0}^{K-1} \frac{\gamma_k}{\gamma} \mathbb{E}_t \|\mathbf{x}_{i,k}^t - \mathbf{x}^t\|^2$$
$$+ 12P\mathbb{E}_t\|\nabla F(\mathbf{z}^t)\|^2 + P(12\sigma_g^2 + \sigma_l^2).$$

### C.4.1 L-SMOOTHNESS OF THE FUNCTION $F$

For the general non-convex case, according to the Assumptions and the smoothness of $F$, we take the conditional expectation at round $t + 1$ and expand the $F(\mathbf{z}^{t+1})$ as:

$$\mathbb{E}_t[F(\mathbf{z}^{t+1})] \leq F(\mathbf{z}^t) + \mathbb{E}_t\langle \nabla F(\mathbf{z}^t), \mathbf{z}^{t+1} - \mathbf{z}^t \rangle + \frac{L}{2}\mathbb{E}_t\|\mathbf{z}^{t+1} - \mathbf{z}^t\|^2$$

$$= F(\mathbf{z}^t) + \langle \nabla F(\mathbf{z}^t), \mathbb{E}_t[\mathbf{z}^{t+1}] - \mathbf{z}^t \rangle + \frac{L}{2}\mathbb{E}_t\|\mathbf{z}^{t+1} - \mathbf{z}^t\|^2$$

$$= F(\mathbf{z}^t) + \mathbb{E}_t\langle \nabla F(\mathbf{z}^t), -\lambda\frac{1}{m} \sum_{i \in [m]} \sum_{k=0}^{K-1} \frac{\gamma_k}{\gamma}\tilde{\mathbf{g}}_{i,k}^t \rangle + \frac{L}{2}\mathbb{E}_t\|\mathbf{z}^{t+1} - \mathbf{z}^t\|^2$$

$$= F(\mathbf{z}^t) - \lambda\mathbb{E}_t\langle \nabla F(\mathbf{z}^t), \frac{1}{m} \sum_{i \in [m]} \sum_{k=0}^{K-1} \frac{\gamma_k}{\gamma}\tilde{\mathbf{g}}_{i,k}^t - \nabla F(\mathbf{z}^t) + \nabla F(\mathbf{z}^t) \rangle$$
$$+ \frac{L}{2}\mathbb{E}_t\|\mathbf{z}^{t+1} - \mathbf{z}^t\|^2$$

$$= F(\mathbf{z}^t) - \lambda\|\nabla F(\mathbf{z}^t)\|^2 \underbrace{-\lambda\mathbb{E}_t\langle \nabla F(\mathbf{z}^t), \frac{1}{m} \sum_{i \in [m]} \sum_{k=0}^{K-1} \frac{\gamma_k}{\gamma}\tilde{\mathbf{g}}_{i,k}^t - \nabla F(\mathbf{z}^t) \rangle}_{\mathbf{R1}}$$

$$+ \frac{L}{2}\underbrace{\mathbb{E}_t\|\mathbf{z}^{t+1} - \mathbf{z}^t\|^2}_{\mathbf{R2}}.$$

### C.4.2 BOUNDED R1

Note that **R1** can be bounded as:

$$\mathbf{R1} = -\lambda\mathbb{E}_t\langle\nabla F(\mathbf{z}^t), \frac{1}{m}\sum_{i\in[m]}\sum_{k=0}^{K-1}\frac{\gamma_k}{\gamma}\tilde{\mathbf{g}}_{i,k}^t - \nabla F(\mathbf{z}^t)\rangle$$

$$\overset{(a)}{=} -\lambda\mathbb{E}_t\langle\nabla F(\mathbf{z}^t), \frac{1}{m}\sum_{i\in[m]}\sum_{k=0}^{K-1}\frac{\gamma_k}{\gamma}\tilde{\mathbf{g}}_{i,k}^t - \frac{1}{m}\sum_{i\in[m]}\sum_{k=0}^{K-1}\frac{\gamma_k}{\gamma}\nabla F_i(\mathbf{z}^t)\rangle$$

$$\overset{(b)}{=} \frac{\lambda}{2}\|\nabla F(\mathbf{z}^t)\|^2 + \frac{\lambda}{2}\mathbb{E}_t\|\frac{1}{m}\sum_{i\in[m]}\sum_{k=0}^{K-1}\frac{\gamma_k}{\gamma}\left(\mathbb{E}\tilde{\mathbf{g}}_{i,k}^t - \nabla F_i(\mathbf{z}^t)\right)\|^2 - \frac{\lambda}{2m^2}\mathbb{E}_t\|\sum_{i\in[m]}\sum_{k=0}^{K-1}\frac{\gamma_k}{\gamma}\mathbb{E}\tilde{\mathbf{g}}_{i,k}^t\|^2$$

$$\overset{(c)}{\leq} \frac{\lambda}{2}\|\nabla F(\mathbf{z}^t)\|^2 + \frac{\lambda}{2}\underbrace{\frac{1}{m}\sum_{i\in[m]}\sum_{k=0}^{K-1}\frac{\gamma_k}{\gamma}\mathbb{E}_t\|\mathbb{E}\tilde{\mathbf{g}}_{i,k}^t - \nabla F_i(\mathbf{z}^t)\|^2}_{\mathbf{R1.a}} - \frac{\lambda}{2m^2}\mathbb{E}_t\|\sum_{i\in[m]}\sum_{k=0}^{K-1}\frac{\gamma_k}{\gamma}\mathbb{E}\tilde{\mathbf{g}}_{i,k}^t\|^2.$$

(a) applies the fact that $\frac{1}{m}\sum_{i\in[m]}\nabla F_i(\mathbf{z}^t) = \nabla F(\mathbf{z}^t)$. (b) applies $-\langle\mathbf{x}, \mathbf{y}\rangle = \frac{1}{2}\left(\|\mathbf{x}\|^2 + \|\mathbf{y}\|^2 - \|\mathbf{x}+\mathbf{y}\|^2\right)$. (c) applies the Jensen's inequality and the fact that $\sum_{k=0}^{K-1}\frac{\gamma_k}{\gamma} = 1$.

According to the update rule we have:

$$\mathbb{E}\tilde{\mathbf{g}}_{i,k}^t = (1-\alpha)\mathbb{E}\left[\mathbf{g}_{i,k,1}^t\right] + \alpha\mathbb{E}\left[\mathbf{g}_{i,k,2}^t\right] = (1-\alpha)\mathbb{E}\left[\nabla F_i(\mathbf{x}_{i,k}^t; \varepsilon_{i,k}^t)\right] + \alpha\mathbb{E}\left[\nabla F_i(\breve{\mathbf{x}}_{i,k}^t; \varepsilon_{i,k}^t)\right]$$

$$= (1-\alpha)\nabla F_i(\mathbf{x}_{i,k}^t) + \alpha\nabla F_i(\breve{\mathbf{x}}_{i,k}^t) = (1-\alpha)\nabla F_i(\mathbf{x}_{i,k}^t) + \alpha\nabla F_i(\mathbf{x}_{i,k}^t + \rho\mathbf{g}_{i,k,1}^t).$$

Let $\rho \leq \frac{1}{\sqrt{3}\alpha L}$, thus we could bound the term **R1.a** as follows:

$$\frac{1}{m}\sum_{i\in[m]}\sum_{k=0}^{K-1}\frac{\gamma_k}{\gamma}\mathbb{E}_t\|\mathbb{E}\tilde{\mathbf{g}}_{i,k}^t - \nabla F_i(\mathbf{z}^t)\|^2$$

$$= \frac{1}{m}\sum_{i\in[m]}\sum_{k=0}^{K-1}\frac{\gamma_k}{\gamma}\mathbb{E}_t\|(1-\alpha)\nabla F_i(\mathbf{x}_{i,k}^t) + \alpha\nabla F_i(\mathbf{x}_{i,k}^t + \rho\mathbf{g}_{i,k,1}^t) - \nabla F_i(\mathbf{z}^t)\|^2$$

$$= \frac{1}{m}\sum_{i\in[m]}\sum_{k=0}^{K-1}\frac{\gamma_k}{\gamma}\mathbb{E}_t\|\nabla F_i(\mathbf{x}_{i,k}^t) - \nabla F_i(\mathbf{z}^t) + \alpha\left(\nabla F_i(\mathbf{x}_{i,k}^t + \rho\mathbf{g}_{i,k,1}^t) - \nabla F_i(\mathbf{x}_{i,k}^t)\right)\|^2$$

$$\leq \frac{2}{m}\sum_{i\in[m]}\sum_{k=0}^{K-1}\frac{\gamma_k}{\gamma}\mathbb{E}_t\|\nabla F_i(\mathbf{x}_{i,k}^t) - \nabla F_i(\mathbf{z}^t)\|^2 + \frac{2\alpha^2}{m}\sum_{i\in[m]}\sum_{k=0}^{K-1}\frac{\gamma_k}{\gamma}\mathbb{E}_t\|\nabla F_i(\breve{\mathbf{x}}_{i,k}^t) - \nabla F_i(\mathbf{x}_{i,k}^t)\|^2$$

$$\leq \frac{2L^2}{m}\sum_{i\in[m]}\sum_{k=0}^{K-1}\frac{\gamma_k}{\gamma}\mathbb{E}_t\|\mathbf{x}_{i,k}^t - \mathbf{z}^t\|^2 + \frac{2\alpha^2 L^2\rho^2}{m}\sum_{i\in[m]}\sum_{k=0}^{K-1}\frac{\gamma_k}{\gamma}\mathbb{E}_t\|\mathbf{g}_{i,k,1}^t\|^2$$

$$= \frac{2L^2}{m}\sum_{i\in[m]}\sum_{k=0}^{K-1}\frac{\gamma_k}{\gamma}\mathbb{E}_t\|\mathbf{x}_{i,k}^t - \mathbf{x}^t + \mathbf{x}^t - \mathbf{u}^t + \mathbf{u}^t - \mathbf{z}^t\|^2 + \frac{2\alpha^2 L^2\rho^2}{m}\sum_{i\in[m]}\sum_{k=0}^{K-1}\frac{\gamma_k}{\gamma}\mathbb{E}_t\|\mathbf{g}_{i,k,1}^t\|^2$$

$$\leq \frac{4L^2}{m}\sum_{i\in[m]}\sum_{k=0}^{K-1}\frac{\gamma_k}{\gamma}\mathbb{E}_t\|\mathbf{x}_{i,k}^t - \mathbf{x}^t\|^2 + \frac{4L^2}{m}\sum_{i\in[m]}\sum_{k=0}^{K-1}\frac{\gamma_k}{\gamma}\mathbb{E}_t\|(\mathbf{x}^t - \mathbf{u}^t) + (\mathbf{u}^t - \mathbf{z}^t)\|^2$$

$$+ \frac{2\alpha^2 L^2\rho^2}{m}\sum_{i\in[m]}\sum_{k=0}^{K-1}\frac{\gamma_k}{\gamma}\mathbb{E}_t\|\mathbf{g}_{i,k,1}^t - \nabla F_i(\mathbf{x}_{i,k}^t)\|^2 + \frac{2\alpha^2 L^2\rho^2}{m}\sum_{i\in[m]}\sum_{k=0}^{K-1}\frac{\gamma_k}{\gamma}\mathbb{E}_t\|\nabla F_i(\mathbf{x}_{i,k}^t)\|^2$$

$$\leq \frac{4L^2}{m}\sum_{i\in[m]}\sum_{k=0}^{K-1}\frac{\gamma_k}{\gamma}\mathbb{E}_t\|\mathbf{x}_{i,k}^t - \mathbf{x}^t\|^2 + \frac{2\alpha^2 L^2\rho^2}{m}\sum_{i\in[m]}\sum_{k=0}^{K-1}\frac{\gamma_k}{\gamma}\mathbb{E}_t\|\nabla F_i(\mathbf{x}_{i,k}^t)\|^2 + 2\alpha^2 L^2\rho^2\sigma_l^2$$

$$+ 4L^2 \mathbb{E}_t \| (\mathbf{x}^t - \mathbf{u}^t) + (\mathbf{u}^t - \mathbf{z}^t) \|^2$$

$$= \frac{4L^2}{m} \sum_{i \in [m]} \sum_{k=0}^{K-1} \frac{\gamma_k}{\gamma} \mathbb{E}_t \| \mathbf{x}_{i,k}^t - \mathbf{x}^t \|^2 + \frac{2\alpha^2 L^2 \rho^2}{m} \sum_{i \in [m]} \sum_{k=0}^{K-1} \frac{\gamma_k}{\gamma} \mathbb{E}_t \| \nabla F_i(\mathbf{x}_{i,k}^t) \|^2 + 2\alpha^2 L^2 \rho^2 \sigma_l^2$$

$$+ 4L^2 \mathbb{E}_t \| - \frac{1}{m} \sum_{i \in [m]} \lambda \hat{\mathbf{g}}_i^{t-1} + \frac{\gamma - 1}{\gamma} (\mathbf{u}^t - \mathbf{u}^{t-1}) \|^2$$

$$= \frac{4L^2}{m} \sum_{i \in [m]} \sum_{k=0}^{K-1} \frac{\gamma_k}{\gamma} \mathbb{E}_t \| \mathbf{x}_{i,k}^t - \mathbf{x}^t \|^2 + \frac{2\alpha^2 L^2 \rho^2}{m} \sum_{i \in [m]} \sum_{k=0}^{K-1} \frac{\gamma_k}{\gamma} \mathbb{E}_t \| \nabla F_i(\mathbf{x}_{i,k}^t) \|^2 + 2\alpha^2 L^2 \rho^2 \sigma_l^2$$

$$+ 4L^2 \mathbb{E}_t \| \frac{1}{m} \sum_{i \in [m]} \left( (\mathbf{u}^t - \mathbf{u}^{t-1} + \lambda \hat{\mathbf{g}}_i^{t-1}) - \frac{1}{\gamma} (\mathbf{u}^t - \mathbf{u}^{t-1} + \lambda \hat{\mathbf{g}}_i^{t-1}) + (\frac{1 - 2\gamma}{\gamma}) \lambda \hat{\mathbf{g}}_i^{t-1} \right) \|^2$$

$$= \frac{4L^2}{m} \sum_{i \in [m]} \sum_{k=0}^{K-1} \frac{\gamma_k}{\gamma} \mathbb{E}_t \| \mathbf{x}_{i,k}^t - \mathbf{x}^t \|^2 + \frac{2\alpha^2 L^2 \rho^2}{m} \sum_{i \in [m]} \sum_{k=0}^{K-1} \frac{\gamma_k}{\gamma} \mathbb{E}_t \| \nabla F_i(\mathbf{x}_{i,k}^t) \|^2 + 2\alpha^2 L^2 \rho^2 \sigma_l^2$$

$$+ \frac{4\lambda^2 L^2 (1 - 2\gamma)^2}{\gamma^2} \mathbb{E}_t \| \frac{1}{m} \sum_{i \in [m]} \hat{\mathbf{g}}_i^{t-1} \|^2$$

$$= \frac{4L^2}{m} \sum_{i \in [m]} \sum_{k=0}^{K-1} \frac{\gamma_k}{\gamma} \mathbb{E}_t \| \mathbf{x}_{i,k}^t - \mathbf{x}^t \|^2 + \frac{4\lambda^2 L^2 (1 - 2\gamma)^2}{\gamma^2} \mathbb{E}_t \| \frac{1}{m} \sum_{i \in [m]} \hat{\mathbf{g}}_i^{t-1} \|^2 + 2\alpha^2 L^2 \rho^2 \sigma_l^2$$

$$+ \frac{2\alpha^2 L^2 \rho^2}{m} \sum_{i \in [m]} \sum_{k=0}^{K-1} \frac{\gamma_k}{\gamma} \mathbb{E}_t \| \nabla F_i(\mathbf{x}_{i,k}^t) - \nabla F_i(\mathbf{z}^t) + \nabla F_i(\mathbf{z}^t) - \nabla F(\mathbf{z}^t) + \nabla F(\mathbf{z}^t) \|^2$$

$$\overset{(a)}{\leq} \frac{4L^2}{m} \sum_{i \in [m]} \sum_{k=0}^{K-1} \frac{\gamma_k}{\gamma} \mathbb{E}_t \| \mathbf{x}_{i,k}^t - \mathbf{x}^t \|^2 + \frac{4\lambda^2 L^2 (1 - 2\gamma)^2}{\gamma^2} \mathbb{E}_t \| \frac{1}{m} \sum_{i \in [m]} \hat{\mathbf{g}}_i^{t-1} \|^2 + 2\alpha^2 L^2 \rho^2 \sigma_l^2$$

$$+ \frac{2L^2}{m} \sum_{i \in [m]} \sum_{k=0}^{K-1} \frac{\gamma_k}{\gamma} \mathbb{E}_t \| \mathbf{x}_{i,k}^t - \mathbf{z}^t \|^2 + 6\alpha^2 L^2 \rho^2 \sigma_g^2 + 6\alpha^2 L^2 \rho^2 \mathbb{E}_t \| \nabla F(\mathbf{z}^t) \|^2$$

$$\leq \frac{8L^2}{m} \sum_{i \in [m]} \sum_{k=0}^{K-1} \frac{\gamma_k}{\gamma} \mathbb{E}_t \| \mathbf{x}_{i,k}^t - \mathbf{x}^t \|^2 + \frac{8\lambda^2 L^2 (1 - 2\gamma)^2}{\gamma^2} \mathbb{E}_t \| \frac{1}{m} \sum_{i \in [m]} \hat{\mathbf{g}}_i^{t-1} \|^2 + 2\alpha^2 L^2 \rho^2 \sigma_l^2$$

$$+ 6\alpha^2 L^2 \rho^2 \sigma_g^2 + 6\alpha^2 L^2 \rho^2 \mathbb{E}_t \| \nabla F(\mathbf{z}^t) \|^2.$$

$$\overset{(b)}{\leq} \frac{8L^2}{m} \sum_{i \in [m]} \sum_{k=0}^{K-1} \frac{\gamma_k}{\gamma} \mathbb{E}_t \| \mathbf{x}_{i,k}^t - \mathbf{x}^t \|^2 + \frac{8\lambda^2 L^2 (1 - 2\gamma)^2}{\gamma^3} \left( \mathbb{E}_t \| \frac{1}{m} \sum_{i \in [m]} \hat{\mathbf{g}}_i^{t-1} \|^2 - \mathbb{E}_t \| \frac{1}{m} \sum_{i \in [m]} \hat{\mathbf{g}}_i^t \|^2 \right)$$

$$+ \frac{8\lambda^2 L^2 (1 - 2\gamma)^2}{\gamma^2} \mathbb{E}_t \| \frac{1}{m} \sum_{i \in [m]} \sum_{k=0}^{K-1} \frac{\gamma_k}{\gamma} \tilde{\mathbf{g}}_{i,k}^t \|^2 + 2\alpha^2 L^2 \rho^2 \sigma_l^2 + 6\alpha^2 L^2 \rho^2 \sigma_g^2 + 6\alpha^2 L^2 \rho^2 \mathbb{E}_t \| \nabla F(\mathbf{z}^t) \|^2.$$

(a) applies the bound of $\rho$ as $\rho \leq \frac{1}{\sqrt{3}\alpha L}$. (b) applies the lemma C.4. These others use the fact $\mathbb{E}[x - \mathbb{E}[x]]^2 = \mathbb{E}[x^2] - [\mathbb{E}[x]]^2$ and $\| \mathbf{x} + \mathbf{y} \|^2 \leq (1 + a) \| \mathbf{x} \|^2 + (1 + \frac{1}{a}) \| \mathbf{y} \|^2$.

We denote $\mathbf{c}^t = \frac{1}{m} \sum_{i \in m} \sum_{k=0}^{K-1} (\gamma_k / \gamma) \mathbb{E}_t \| \mathbf{x}_{i,k}^t - \mathbf{x}^t \|^2$ term as the local offset after $k$ iterations updates, we firstly consider the $\mathbf{c}_k^t = \frac{1}{m} \sum_{i \in m} \mathbb{E}_t \| \mathbf{x}_{i,k}^t - \mathbf{x}^t \|^2$ and it can be bounded as:

$$\mathbf{c}_k^t = \frac{1}{m} \sum_{i \in [m]} \mathbb{E}_t \| \mathbf{x}_{i,k}^t - \mathbf{x}^t \|^2 = \frac{1}{m} \sum_{i \in [m]} \mathbb{E}_t \| \mathbf{x}_{i,k}^t - \mathbf{x}_{i,k-1}^t + \mathbf{x}_{i,k-1}^t - \mathbf{x}_{i,0}^t \|^2$$

$$= \frac{1}{m} \sum_{i \in [m]} \mathbb{E}_t \| - \eta_l (\tilde{\mathbf{g}}_{i,k-1}^t - \hat{\mathbf{g}}_i^{t-1}) + (1 - \frac{\eta_l}{\lambda})(\mathbf{x}_{i,k-1}^t - \mathbf{x}_{i,0}^t) \|^2$$

$$\leq (1+a)(1-\frac{\eta_l}{\lambda})^2\frac{1}{m}\sum_{i\in[m]}\mathbb{E}_t\|\mathbf{x}_{i,k-1}^t - \mathbf{x}_{i,0}^t\|^2 + (1+\frac{1}{a})\frac{\eta_l^2}{m}\sum_{i\in[m]}\mathbb{E}_t\|\tilde{\mathbf{g}}_{i,k-1}^t - \hat{\mathbf{g}}_i^{t-1}\|^2$$

$$= (1+a)(1-\frac{\eta_l}{\lambda})^2\mathbf{c}_{k-1}^t + (1+\frac{1}{a})\frac{\eta_l^2}{m}\sum_{i\in[m]}\mathbb{E}_t\|(1-\alpha)\mathbf{g}_{i,k-1,1}^t + \alpha\mathbf{g}_{i,k-1,2}^t - \hat{\mathbf{g}}_i^{t-1}\|^2$$

$$= (1+\frac{1}{a})\frac{\eta_l^2}{m}\sum_{i\in[m]}\mathbb{E}_t\|\nabla F_i(\mathbf{x}_{i,k-1}^t) - \hat{\mathbf{g}}_i^{t-1} + \alpha(\nabla F_i(\breve{\mathbf{x}}_{i,k-1}^t) - \nabla F_i(\mathbf{x}_{i,k-1}^t))\|^2$$

$$+ (1+\frac{1}{a})\eta_l^2\sigma_l^2 + (1+a)(1-\frac{\eta_l}{\lambda})^2\mathbf{c}_{k-1}^t$$

$$\leq (1+\frac{1}{a})\frac{3\eta_l^2}{m}\sum_{i\in[m]}\left(\mathbb{E}_t\|\nabla F_i(\mathbf{x}_{i,k-1}^t)\|^2 + \mathbb{E}_t\|\hat{\mathbf{g}}_i^{t-1}\|^2 + \alpha^2 L^2\rho^2\mathbb{E}_t\|\nabla F_i(\mathbf{x}_{i,k-1}^t)\|^2\right)$$

$$+ (1+\frac{1}{a})\eta_l^2\sigma_l^2 + (1+a)(1-\frac{\eta_l}{\lambda})^2\mathbf{c}_{k-1}^t$$

$$\leq (1+\frac{1}{a})\frac{4\eta_l^2}{m}\sum_{i\in[m]}\mathbb{E}_t\|\nabla F_i(\mathbf{x}_{i,k-1}^t)\|^2 + (1+\frac{1}{a})\frac{3\eta_l^2}{m}\sum_{i\in[m]}\mathbb{E}_t\|\hat{\mathbf{g}}_i^{t-1}\|^2 + (1+\frac{1}{a})\eta_l^2\sigma_l^2$$

$$+ (1+a)(1-\frac{\eta_l}{\lambda})^2\mathbf{c}_{k-1}^t$$

$$\leq (1+\frac{1}{a})\frac{4\eta_l^2}{m}\sum_{i\in[m]}\mathbb{E}_t\|\nabla F_i(\mathbf{x}_{i,k-1}^t) - \nabla F_i(\mathbf{x}^t) + \nabla F_i(\mathbf{x}^t) - \nabla F_i(\mathbf{z}^t) + \nabla F_i(\mathbf{z}^t) - \nabla F(\mathbf{z}^t)$$

$$+ \nabla F(\mathbf{z}^t)\|^2 + (1+\frac{1}{a})\frac{3\eta_l^2}{m}\sum_{i\in[m]}\mathbb{E}_t\|\hat{\mathbf{g}}_i^{t-1}\|^2 + (1+\frac{1}{a})\eta_l^2\sigma_l^2 + (1+a)(1-\frac{\eta_l}{\lambda})^2\mathbf{c}_{k-1}^t$$

$$\leq (1+\frac{1}{a})\frac{16\eta_l^2 L^2}{m}\sum_{i\in[m]}\mathbb{E}_t\|\mathbf{x}_{i,k-1}^t - \mathbf{x}^t\|^2 + (1+\frac{1}{a})16\eta_l^2 L^2\|\mathbf{x}^t - \mathbf{z}^t\|^2 + (1+\frac{1}{a})\eta_l^2(16\sigma_g^2 + \sigma_l^2)$$

$$+ (1+\frac{1}{a})16\eta_l^2\|\nabla F(\mathbf{z}^t)\|^2 + (1+\frac{1}{a})\frac{3\eta_l^2}{m}\sum_{i\in[m]}\mathbb{E}_t\|\hat{\mathbf{g}}_i^{t-1}\|^2 + (1+a)(1-\frac{\eta_l}{\lambda})^2\mathbf{c}_{k-1}^t$$

$$\leq \left[(1+a)(1-\frac{\eta_l}{\lambda})^2 + (1+\frac{1}{a})16\eta_l^2 L^2\right]\mathbf{c}_{k-1}^t + (1+\frac{1}{a})\eta_l^2(16\sigma_g^2 + \sigma_l^2)$$

$$+ (1+\frac{1}{a})16\eta_l^2\mathbb{E}_t\|\nabla F(\mathbf{z}^t)\|^2 + (1+\frac{1}{a})\eta_l^2\left[3 + \frac{16\lambda^2 L^2(1-2\gamma)^2}{\gamma^2}\right]\frac{1}{m}\sum_{i\in[m]}\mathbb{E}_t\|\hat{\mathbf{g}}_i^{t-1}\|^2$$

$$= \left[(1+a)(1-\frac{\eta_l}{\lambda})^2 + (1+\frac{1}{a})16\eta_l^2 L^2\right]\mathbf{c}_{k-1}^t + (1+\frac{1}{a})\eta_l^2(16\sigma_g^2 + \sigma_l^2)$$

$$+ (1+\frac{1}{a})\eta_l^2 L^2(88P - 16)\mathbf{c}^t + (1+\frac{1}{a})\frac{2\eta_l^2(P-1)}{3}(12\sigma_g^2 + \sigma_l^2)$$

$$+ (1+\frac{1}{a})16\eta_l^2\mathbb{E}_t\|\nabla F(\mathbf{z}^t)\|^2 + (1+\frac{1}{a})\eta_l^2(44P - 8)\mathbb{E}_t\|\nabla F(\mathbf{z}^t)\|^2$$

$$+ (1+\frac{1}{a})\frac{2\eta_l^2(P-1)}{3\gamma}\frac{1}{m}\sum_{i\in[m]}\left(\mathbb{E}_t\|\hat{\mathbf{g}}_i^{t-1}\|^2 - \mathbb{E}_t\|\hat{\mathbf{g}}_i^t\|^2\right)$$

When $P$ satisfies the condition of $P \leq 2$, which means $\frac{1}{P} = 1 - \frac{24\lambda^2 L^2(1-2\gamma)^2}{\gamma^2} \geq \frac{1}{2}$, then we have the constant of $\frac{2(P-1)}{3} \leq \frac{2}{3} < 1$, let the last $12\sigma_g^2$ enlarged to $16\sigma_g^2$ for convenience, we have:

$$\mathbf{c}_k^t \leq \left[(1+a)(1-\frac{\eta_l}{\lambda})^2 + (1+\frac{1}{a})16\eta_l^2 L^2\right]\mathbf{c}_{k-1}^t + 2(1+\frac{1}{a})\eta_l^2(16\sigma_g^2 + \sigma_l^2) + 160(1+\frac{1}{a})\eta_l^2 L^2\mathbf{c}^t$$

$$96(1 + \frac{1}{a})\eta_l^2 \mathbb{E}_t \|\nabla F(\mathbf{z}^t)\|^2 + 2(1 + \frac{1}{a})\frac{\eta_l^2}{\gamma} \frac{1}{m} \sum_{i \in [m]} \left( \mathbb{E}_t \|\hat{\mathbf{g}}_i^{t-1}\|^2 - \mathbb{E}_t \|\hat{\mathbf{g}}_i^t\|^2 \right).$$

Here we get the recursion formula between the $\mathbf{c}_k^t$ and $\mathbf{c}_{k-1}^t$. Actually we need to upper bound the $\mathbf{c}^t = \sum_{k=0}^{K-1} (\gamma_k / \gamma) \mathbf{c}_k^t$, thus let the weight satisfies that:

$$(1 + a)(1 - \frac{\eta_l}{\lambda})^2 + (1 + \frac{1}{a})16\eta_l^2 L^2 \le \frac{\gamma_{K-2}}{\gamma_{K-1}} = \frac{\gamma_{K-3}}{\gamma_{K-2}} = \cdots = \frac{\gamma_1}{\gamma_0} = 1 - \frac{\eta_l}{\lambda},$$

let $\eta_l \le \lambda$ and thus we have:

$$\mathbf{c}^t = \sum_{k=0}^{K-1} \frac{\gamma_k}{\gamma} \mathbf{c}_k^t$$

$$\le 2(1 + \frac{1}{a})\frac{\eta_l^2}{\gamma} \sum_{k'=0}^{K-1} \left( \sum_{k=0}^{k'-1} \gamma_k \right) \left( 16\sigma_g^2 + \sigma_l^2 + 48\mathbb{E}_t\|\nabla F(\mathbf{z}^t)\|^2 + 80L^2\mathbf{c}^t \right.$$

$$+ \frac{1}{m\gamma} \sum_{i \in [m]} \left( \mathbb{E}_t\|\hat{\mathbf{g}}_i^{t-1}\|^2 - \mathbb{E}_t\|\hat{\mathbf{g}}_i^t\|^2 \right) \Big)$$

$$\overset{(a)}{\le} 2(1 + \frac{1}{a})\eta_l^2 \sum_{k'=0}^{K-1} \left( \sum_{k=0}^{K-1} \frac{\gamma_k}{\gamma} \right) \left( 16\sigma_g^2 + \sigma_l^2 + 48\mathbb{E}_t\|\nabla F(\mathbf{z}^t)\|^2 + 80L^2\mathbf{c}^t \right.$$

$$+ \frac{1}{m\gamma} \sum_{i \in [m]} \left( \mathbb{E}_t\|\hat{\mathbf{g}}_i^{t-1}\|^2 - \mathbb{E}_t\|\hat{\mathbf{g}}_i^t\|^2 \right) \Big)$$

$$= 2(1 + \frac{1}{a})\eta_l^2 K \left( 16\sigma_g^2 + \sigma_l^2 + 48\mathbb{E}_t\|\nabla F(\mathbf{z}^t)\|^2 + \frac{1}{m\gamma} \sum_{i \in [m]} \left( \mathbb{E}_t\|\hat{\mathbf{g}}_i^{t-1}\|^2 - \mathbb{E}_t\|\hat{\mathbf{g}}_i^t\|^2 \right) \right)$$

$$+ 160(1 + \frac{1}{a})\eta_l^2 L^2 K \mathbf{c}^t.$$

(a) enlarge the sum from $k'$ to $K - 1$ where $k' \le K - 1$.
Let $\eta_l$ satisfies the upper bound of $\eta_l \le \frac{1}{\sqrt{320(1+1/a)KL}}$ for convenience, we can bound the $\mathbf{c}^t$ as:

$$\mathbf{c}^t = 4(1 + \frac{1}{a})\eta_l^2 K \left( 16\sigma_g^2 + \sigma_l^2 + 48\mathbb{E}_t\|\nabla F(\mathbf{z}^t)\|^2 + \frac{1}{m\gamma} \sum_{i \in [m]} \left( \mathbb{E}_t\|\hat{\mathbf{g}}_i^{t-1}\|^2 - \mathbb{E}_t\|\hat{\mathbf{g}}_i^t\|^2 \right) \right).$$

Let the $a$ satisfies $a = 1$ for convenience, we summarize the extra terms above and bound the term **R1.a** as:

$$\mathbf{R1.a} = \frac{1}{m} \sum_{i \in [m]} \sum_{k=0}^{K-1} \frac{\gamma_k}{\gamma} \mathbb{E}_t \|\mathbb{E}[\tilde{\mathbf{g}}_{i,k}^t] - \nabla F_i(\mathbf{z}^t)\|^2$$

$$\le 8L^2\mathbf{c}^t + \frac{8\lambda^2 L^2(1-2\gamma)^2}{\gamma^3} \left( \mathbb{E}_t\|\frac{1}{m}\sum_{i \in [m]}\hat{\mathbf{g}}_i^{t-1}\|^2 - \mathbb{E}_t\|\frac{1}{m}\sum_{i \in [m]}\hat{\mathbf{g}}_i^t\|^2 \right) + 2\alpha^2 L^2\rho^2\sigma_l^2$$

$$+ \frac{8\lambda^2 L^2(1-2\gamma)^2}{\gamma^2}\mathbb{E}_t\|\frac{1}{m}\sum_{i \in [m]}\sum_{k=0}^{K-1}\frac{\gamma_k}{\gamma}\tilde{\mathbf{g}}_{i,k}^t\|^2 + 6\alpha^2 L^2\rho^2\sigma_g^2 + 6\alpha^2 L^2\rho^2\mathbb{E}_t\|\nabla F(\mathbf{z}^t)\|^2$$

$$\le \frac{8\lambda^2 L^2(1-2\gamma)^2}{\gamma^3} \left( \mathbb{E}_t\|\frac{1}{m}\sum_{i \in [m]}\hat{\mathbf{g}}_i^{t-1}\|^2 - \mathbb{E}_t\|\frac{1}{m}\sum_{i \in [m]}\hat{\mathbf{g}}_i^t\|^2 \right) + 2\alpha^2 L^2\rho^2\sigma_l^2 + 6\alpha^2 L^2\rho^2\sigma_g^2$$

$$+ \frac{8\lambda^2 L^2 (1-2\gamma)^2}{\gamma^2} \mathbb{E}_t \| \frac{1}{m} \sum_{i\in[m]} \sum_{k=0}^{K-1} \frac{\gamma_k}{\gamma} \tilde{\mathbf{g}}_{i,k}^t \|^2 + \frac{64\eta_l^2 L^2 K}{m\gamma} \sum_{i\in[m]} \left( \mathbb{E}_t \|\hat{\mathbf{g}}_i^{t-1}\|^2 - \mathbb{E}_t \|\hat{\mathbf{g}}_i^t\|^2 \right)$$

$$+ 3072\eta_l^2 L^2 K \mathbb{E}_t \|\nabla F(\mathbf{z}^t)\|^2 + 6\alpha^2 L^2 \rho^2 \mathbb{E}_t \|\nabla F(\mathbf{z}^t)\|^2 + 64\eta_l^2 L^2 K (16\sigma_g^2 + \sigma_l^2).$$

thus we can bound the **R1** as follow:

$$\mathbf{R1} \leq \frac{\lambda}{2} \mathbb{E}_t \|\nabla F(\mathbf{z}^t)\|^2 + \frac{\lambda}{2} \mathbf{R1.a} - \frac{\lambda}{2m^2} \mathbb{E}_t \| \sum_{i\in[m]} \sum_k^{K-1} \frac{\gamma_k}{\gamma} \mathbb{E}[\tilde{\mathbf{g}}_{i,k}^t] \|^2$$

$$\leq \left( \frac{\lambda}{2} + 3\lambda\alpha^2 L^2 \rho^2 + 1536\lambda\eta_l^2 L^2 K \right) \mathbb{E}_t \|\nabla F(\mathbf{z}^t)\|^2 + \frac{32\lambda\eta_l L^2 K}{\gamma m} \sum_{i\in[m]} \left( \mathbb{E}\|\hat{\mathbf{g}}_i^{t-1}\|^2 - \mathbb{E}\|\hat{\mathbf{g}}_i^t\|^2 \right)$$

$$+ \frac{4\lambda^3 L^2 (1-2\gamma)^2}{\gamma^3} \left( \mathbb{E}_t \| \frac{1}{m} \sum_{i\in[m]} \hat{\mathbf{g}}_i^{t-1} \|^2 - \mathbb{E}_t \| \frac{1}{m} \sum_{i\in[m]} \hat{\mathbf{g}}_i^t \|^2 \right) + \lambda\alpha^2 L^2 \rho^2 (3\sigma_g^2 + \sigma_l^2)$$

$$+ \frac{4\lambda^3 L^2 (1-2\gamma)^2}{\gamma^2} \mathbb{E}_t \| \frac{1}{m} \sum_{i\in[m]} \sum_{k=0}^{K-1} \frac{\gamma_k}{\gamma} \tilde{\mathbf{g}}_{i,k}^t \|^2 + 32\lambda\eta_l^2 L^2 K (16\sigma_g^2 + \sigma_l^2).$$

We notice that **R1** contains the same term with a negative weight, thus we can set another constrains for $\lambda$ to eliminate this term. We will prove it in the next part.

### C.4.3 BOUNDED GLOBAL GRADIENT

As we have bounded the term **R1** and **R2**, according to the smoothness inequality, we combine the inequalities above and get the inequality:

$$\mathbb{E}_t[F(\mathbf{z}^{t+1})] \leq F(\mathbf{z}^t) - \lambda\|\nabla F(\mathbf{z}^t)\|^2 + \mathbf{R1} + \frac{L}{2}\mathbf{R2}$$

$$= F(\mathbf{z}^t) - \left( \frac{\lambda}{2} - 3\lambda\alpha^2 L^2 \rho^2 - 1536\lambda\eta_l^2 L^2 K \right) \|\nabla F(\mathbf{z}^t)\|^2 + \lambda\alpha^2 L^2 \rho^2 (3\sigma_g^2 + \sigma_l^2)$$

$$+ \left( \frac{4\lambda^3 L^2 (1-2\gamma)^2}{\gamma^2} + \frac{\lambda^2 L}{2m^2} - \frac{\lambda}{2m^2} \right) \mathbb{E}_t \| \sum_{i\in[m]} \sum_{k=0}^{K-1} \frac{\gamma_k}{\gamma} \tilde{\mathbf{g}}_{i,k}^t \|^2$$

$$+ \frac{32\lambda\eta_l L^2 K}{\gamma m} \sum_{i\in[m]} \left( \mathbb{E}\|\hat{\mathbf{g}}_i^{t-1}\|^2 - \mathbb{E}\|\hat{\mathbf{g}}_i^t\|^2 \right) + 32\lambda\eta_l^2 L^2 K (16\sigma_g^2 + \sigma_l^2)$$

$$+ \frac{4\lambda^3 L^2 (1-2\gamma)^2}{\gamma^3} \left( \mathbb{E}_t \| \frac{1}{m} \sum_{i\in[m]} \hat{\mathbf{g}}_i^{t-1} \|^2 - \mathbb{E}_t \| \frac{1}{m} \sum_{i\in[m]} \hat{\mathbf{g}}_i^t \|^2 \right).$$

We follow as Yang et al. (2021) to set $\lambda$ that it satisfies $\frac{4\lambda^3 L^2 (1-2\gamma)^2}{\gamma^2} + \frac{\lambda^2 L}{2m^2} - \frac{\lambda}{2m^2} \leq 0$, which is easy to verified that $\lambda$ has a upper bound for the quadratic inequality. Thus, the stochastic gradient term is diminished by this $\lambda$. We denote the constant $\lambda\kappa = \frac{\lambda}{2} - 3\lambda\alpha^2 L^2 \rho^2 - 1536\lambda\eta_l^2 L^2 K$ and $\kappa$ could be considered as a constant. We can select two constants $c_1 \in (0, \frac{1}{2}), c_2 \in (0, \frac{1}{2})$ and they satisfy $c_1 + c_2 \in (0, \frac{1}{2})$, we let $\frac{1}{2} - 3\alpha^2 L^2 \rho^2 > \frac{1}{2} - c_1$ and $\frac{1}{2} - 1536\eta_l^2 L^2 K > \frac{1}{2} - c_2$, where the $\rho$ and $\eta_l$ satisfy $\rho < \frac{\sqrt{c_1}}{\sqrt{3}\alpha L} < \frac{1}{\sqrt{6}\alpha L}$ and $\eta_l < \frac{\sqrt{c_2}}{16\sqrt{6K}L} < \frac{1}{32\sqrt{3K}L}$. Then we can bound the $\kappa = \frac{1}{2} - 3\alpha^2 L^2 \rho^2 - 1536\eta_l^2 L^2 K > \frac{1}{2} - c_1 - c_2 > 0$, and the term $\frac{1}{\kappa} < \frac{2}{1-2c_1-2c_2}$ which is a constant upper bound.

We take the full expectation on the bounded global gradient as:

$$\lambda\kappa\mathbb{E}\|\nabla F(\mathbf{z}^t)\|^2 \leq \left( \mathbb{E}F(\mathbf{z}^t) - \mathbb{E}F(\mathbf{z}^{t+1}) \right) + \frac{32\lambda\eta_l L^2 K}{\gamma m} \sum_{i\in[m]} \left( \mathbb{E}\|\hat{\mathbf{g}}_i^{t-1}\|^2 - \mathbb{E}\|\hat{\mathbf{g}}_i^t\|^2 \right)$$

$$+ \frac{4\lambda^3 L^2 (1-2\gamma)^2}{\gamma^3} \left( \mathbb{E}_t \| \frac{1}{m} \sum_{i \in [m]} \hat{\mathbf{g}}_i^{t-1} \|^2 - \mathbb{E}_t \| \frac{1}{m} \sum_{i \in [m]} \hat{\mathbf{g}}_i^t \|^2 \right)$$

$$+ 32\lambda\eta_l^2 L^2 K (16\sigma_g^2 + \sigma_l^2) + \lambda\alpha^2 L^2 \rho^2 (3\sigma_g^2 + \sigma_l^2).$$

Take the full expectation and telescope sum on the inequality above and applying the fact that $F^* \leq F(\mathbf{x})$ for $\mathbf{x} \in \mathbb{R}^d$, we have:

$$\frac{1}{T} \sum_{t=1}^{T-1} \mathbb{E}_t \|\nabla F(\mathbf{z}^t)\|^2 \leq \frac{1}{\lambda\kappa T} \left( F(\mathbf{z}^1) - \mathbb{E}_t[F(\mathbf{z}^T)] \right) + \frac{32\eta_l L^2 K}{\kappa\gamma m T} \sum_{i \in [m]} \left( \mathbb{E}\|\hat{\mathbf{g}}_i^0\|^2 - \mathbb{E}\|\hat{\mathbf{g}}_i^t\|^2 \right)$$

$$+ \frac{4\lambda^2 L^2 (1-2\gamma)^2}{\kappa\gamma^3 T} \left( \mathbb{E}_t \| \frac{1}{m} \sum_{i \in [m]} \hat{\mathbf{g}}_i^0 \|^2 - \mathbb{E}_t \| \frac{1}{m} \sum_{i \in [m]} \hat{\mathbf{g}}_i^t \|^2 \right)$$

$$+ \frac{1}{\kappa} \left( 32\lambda\eta_l^2 L^2 K (16\sigma_g^2 + \sigma_l^2) + \lambda\alpha^2 L^2 \rho^2 (3\sigma_g^2 + \sigma_l^2) \right)$$

$$\leq \frac{1}{\lambda\kappa T} \left( F(\mathbf{z}^0) - F^* \right) + \frac{32\eta_l L^2 K}{\kappa\gamma m T} \sum_{i \in [m]} \mathbb{E}\|\hat{\mathbf{g}}_i^0\|^2$$

$$+ \frac{4\lambda^2 L^2 (1-2\gamma)^2}{\kappa\gamma^3 T} \mathbb{E}_t \| \frac{1}{m} \sum_{i \in [m]} \hat{\mathbf{g}}_i^0 \|^2$$

$$+ \frac{1}{\kappa} \left( 32\lambda\eta_l^2 L^2 K (16\sigma_g^2 + \sigma_l^2) + \lambda\alpha^2 L^2 \rho^2 (3\sigma_g^2 + \sigma_l^2) \right)$$

Here we summarize the conditions and some constrains in the above conclusion. Firstly we should note that $\gamma = 1 - (1 - \frac{\eta_l}{\lambda})^K < 1$ when $\eta_l \leq 2\lambda$. Thus we have $1/\gamma > 1$. When $K$ satisfies that $K \geq \frac{\lambda}{\eta_l}$, $(1 - \frac{\eta_l}{\lambda})^K \leq e^{-\frac{\eta_l}{\lambda} K} \leq e^{-1}$, and then $\gamma > 1 - e^{-1}$ and $1/\gamma < \frac{e}{e-1} < 2$. To let $\kappa = \frac{1}{2} - 3\alpha^2 L^2 \rho^2 - 1536\eta_l^2 L^2 K > 0$ hold, $\rho$ and $\eta_l$ satisfy that $\rho < \frac{1}{\sqrt{6}\alpha L}$ and $\eta_l < \frac{1}{32\sqrt{3KL}}$.

$$\frac{1}{T} \sum_{t=1}^{T-1} \mathbb{E}\|\nabla F(\mathbf{z}^t)\|^2 \leq \frac{2(F(\mathbf{z}^1) - F^*)}{\lambda\kappa T} + \frac{64\eta_l L^2 K}{\kappa T} \frac{1}{m} \sum_{i \in [m]} \mathbb{E}\|\hat{\mathbf{g}}_i^0\|^2 + \frac{32\lambda^2 L^2}{\kappa T} \mathbb{E}_t \| \frac{1}{m} \sum_{i \in [m]} \hat{\mathbf{g}}_i^0 \|^2$$

$$+ \frac{1}{\kappa} \left( 32\lambda\eta_l^2 L^2 K (16\sigma_g^2 + \sigma_l^2) + \lambda\alpha^2 L^2 \rho^2 (3\sigma_g^2 + \sigma_l^2) \right).$$

