# OpenReview forum: "FedSpeed: Larger Local Interval, Less Communication Round, and Higher Generalization Accuracy"
_ICLR.cc/2023/Conference — ICLR 2023 poster_

### Official Review · Reviewer_W3wF · 2022-10-26

**Confidence:** 4
**Correctness:** 4
**Technical Novelty And Significance:** 3
**Empirical Novelty And Significance:** 3
**Recommendation:** 6

**Clarity, Quality, Novelty And Reproducibility:**

Paper is well-written.


**Strength And Weaknesses:**

Strength:

The experiment section is convincing to me. The way I see it, roughly speaking this paper uses both “gradient tracking” with additional “regularization” to deal with hetregenoty and improve generalization error which is interesting to me.

Comments:

There is no improvement in terms of achieved rates.
As mentioned on page 4, prox-term introduces an additional bias term. So, my question is do you show this extra gradient term can help to remove that bias?
In the title, it is mentioned “Higher Generalization error”, yet there is no theoretical proof for this claim (just verified through some experiments).
Authors keep mentioning that having larger local updates is a good property. However, there are a couple of papers such as [B] where it is suggested that there is no clear benefit to having a bigger local step size in FL, which contradicts the statement of this paper.
Minor:

The following references [C] and [D] are using gradient tracking ideas (which show improvement over SCAFFOLD) so I would recommend authors compare their results with these references theoretically and empirically.

[B]: Woodworth, Blake E., Kumar Kshitij Patel, and Nati Srebro. "Minibatch vs local sgd for heterogeneous distributed learning." Advances in Neural Information Processing Systems 33 (2020): 6281-6292.

[C]: X. Liang, S. Shen, J. Liu, Z. Pan, E. Chen, and Y. Cheng. Variance reduced local sgd with lower communication complexity. arXiv preprint arXiv:1912.12844, 2019.

[D]: Haddadpour, Farzin, Mohammad Mahdi Kamani, Aryan Mokhtari, and Mehrdad Mahdavi. "Federated learning with compression: Unified analysis and sharp guarantees." In International Conference on Artificial Intelligence and Statistics, pp. 2350-2358. PMLR, 2021.


**Summary Of The Paper:**

This paper proposes FedSpeed algorithm which has two components 1) prox-correction term and 2) extragradient perturbation where they relax a couple of assumptions made in prior work while achieving the same rate.



**Summary Of The Review:**

Please see the comments above!
I will update my score based on the authors feedback.

---

> ### Author Response · Authors · 2022-11-07
> **Response to Reviewer W3wF (Part 3/3):**
>
>
> ## 5. About the question of "compare the results with the references [C] and [D] in the review":
> Thanks a lot to point out these two works. We will introduce these two works in the section on "related works".
>
> [D] proposes to utilize compression during the communication and the server aggregates the compressed information to update while the local clients use the compressed tracking gradient to update. However, FedSpeed does not focus on the compression technique. In fact, FedSpeed and the baselines in our paper are difficult to compare fairly with [D]. Theoretical analysis of the compression techniques usually relies on additional assumptions, such as Assumption 2 in [D], to upper bound the compressed variable. It is usually difficult to assess whether this assumption does not conflict with the common assumptions or which is much stronger. In practical training, its performance highly relies on the compression ratio. We will add this work to the "Related works".
>
> [C] propose a variance reduction local SGD method, which can be regarded as similar to SVRG and SAGA. The key idea is also similar to the SCAFFOLD to adopt the variance reduction technique under the full participation case. However, the local SGD method requires that the whole clients participate in the training per round. While FL usually allows partial participation case in the training. The proposed VAR-SGD algorithm can not be directly extended to partial participation setups because the local tracking gradient must update each round. In fact, FL is more difficult than the local SGD method due to partial participation. [7] has proven that if FedAvg changes the partial participation to the full participation (Local-SGD-type), the dominant term of convergence rate will change from $\mathcal{O}(\frac{\sqrt{K}}{\sqrt{nT}})$ to $\mathcal{O}(\frac{1}{\sqrt{mKT}})$, which will be relaxed to $K$ times faster. Full participation usually achieves a higher theoretical rate than partial participation. In our analysis, FedSpeed achieves $\mathcal{O}(\frac{1}{T})$ which is still faster than the VRL-SGD in [C]. VRL-SGD can theoretically improve the efficiency by adopting a larger order of local interval $K=\mathcal{O}(T^{\frac{1}{2}})$ than FedAvg, while FedSpeed can adopt $K=\mathcal{O}(T)$. Extensive experiments are conducted to verify that FedSpeed works well in practical training which matches its theoretical performance. We will add this work in the "Related works" and discuss it in the appendix.
>
>
> [C] X. Liang, S. Shen, J. Liu, Z. Pan, E. Chen, and Y. Cheng. Variance reduced local sgd with lower communication complexity. arXiv preprint arXiv:1912.12844, 2019.
>
> [D] Haddadpour, Farzin, Mohammad Mahdi Kamani, Aryan Mokhtari, and Mehrdad Mahdavi. "Federated learning with compression: Unified analysis and sharp guarantees." In International Conference on Artificial Intelligence and Statistics, pp. 2350-2358. PMLR, 2021.
>
> [7] Yang H, Fang M, Liu J. Achieving linear speedup with partial worker participation in non-IID federated learning. ICLR 2021.
>
> #### It is a pleasure to discuss this with you, which will help us to further improve this work. We explain and prove the concerns mentioned in the reviews. If there are any questions, we are happy to continue the discussion with you. Thank you again for reading this rebuttal.

---

> ### Author Response · Authors · 2022-11-07
> **Response to Reviewer W3wF  (Part 2/3):**
>
> ## 3. About the question of "do you show the extra gradient term can help to remove the bias?":
> As we mentioned in Section 3.1 paragraph "Gradient perturbation", FedSpeed adopts the extra ascent step gradient as a gradient perturbation to significantly improve the generalization accuracy in the practical training. The main insights and explanations are stated in answer 2 above. We check the paper again and do not find the statement "this extra gradient term can help to remove that bias". If there are similar expressions, please mark them and we will revise the description.
>
> In Section 3.1 paragraph "Prox-correction term", FedSpeed adopts the prox-correction term to further strengthen the consistency during the local training. [1] indicate that this regularization term will introduce non-vanishing inconsistent biases in the local training, which implies a large local inconsistency. Thus FedSpeed uses the correction term to reduce the negative impact. In [2] they bound the averaged l2-norm of the local difference $\frac{1}{m}\sum_{i}\Vert x_{i,k}^{t}-x_{t}\Vert^{2}$, in which the constant bound is $\mathcal{O}(K\eta_{l}^{2}\sigma_{l}^{2}+K^{2}\eta_{l}^{2}\sigma_{g}^{2})$. We bound the similar term $\mathcal{c}$ in our proof of Appendix (page.23 to page.25). In our proof, the constant part could achieve the $\mathcal{O}(K\eta_{l}^{2}\sigma_{l}^{2}+K\eta_{l}^{2}\sigma_{g}^{2})$. This demonstrates if the dominant term comes from the heterogeneous variance $\sigma_{g}^{2}$ (e.g. $\sigma_{g}^{2}$ is far larger than $\sigma_{l}^{2}$), FedSpeed can bound the $\mathcal{O}(K)\times$ smaller local difference than before.
>
> [1] Filip Hanzely and Peter Richtarik. Federated learning of a mixture of global and local models.
>
> [2] Yang H, Fang M, Liu J. Achieving linear speedup with partial worker participation in non-IID federated learning. (ICLR 2021).
>
> ## 4. About the question of "larger local updates is a good property which is different from the statement of [2]":
>
> In [2] they state that in the local SGD method, increasing local interval $K$ will not improve the performance strictly, especially compared with the minibatch SGD method the local SGD can not achieve a faster rate. This is also proven in [3,4] that FedAvg can not benefit from increasing the local interval $K$ directly. The main reason is mentioned as the ($\frac{\sqrt{K}}{\sqrt{nT}}$) term, which means that Increasing $K$ does not necessarily increase the rate.
>
> In our paper, what we claim is that if a federated learning method could adopt a large $K$, it is a good property. Unfortunately, most SGD-type algorithms cannot improve performance by increasing $K$. For example, FedAvg and FedCM require the $K$ as a constant bound. Some useful techniques are adopted in the FL framework to improve the performance, e.g. variance reduction, gradient tracking, and regularization term (mainly the prox-based methods). In the paper [C], VAR-SGD could adopt a larger $K$ of $\mathcal{O}(T^{\frac{1}{2}})$ order.
>
> FedSpeed is a novel prox-based method that incorporates the correction term and extra ascent gradient to improve performance. In fact, in [5,6], it has been proven that prox-based methods have the potential to apply the larger local interval in the local training under the requirement of local minimal solution per communication round. We theoretically prove that FedSpeed can achieve the fast rate without this harsh assumption and it can apply the large $K$ in the local client. We conduct extensive experiments to verify its efficiency. In Figure.2 of our paper, we test the performance of increasing local interval and decreasing the communication rounds under the fixed sample complexity, which shows that FedSpeed is useful in practical training.
>
> In summary, our statement does not contradict the previous conclusion. Under the condition of ensuring the convergence rate, it is one of the most desirable goals for FL to use the large local interval instead of the requirements of the communication rounds, which can efficiently reduce communication costs.
>
>
> [C]: X. Liang, S. Shen, J. Liu, Z. Pan, E. Chen, and Y. Cheng. Variance reduced local sgd with lower communication complexity.
>
> [2] Woodworth, Blake E., Kumar Kshitij Patel, and Nati Srebro. "Minibatch vs local sgd for heterogeneous distributed learning." Advances in Neural Information Processing Systems 33 (2020): 6281-6292.
>
> [3] Yang H, Fang M, Liu J. Achieving linear speedup with partial worker participation in non-IID federated learning. (ICLR 2021).
>
> [4] Karimireddy S P, Kale S, Mohri M, et al. Scaffold: Stochastic controlled averaging for federated learning. International Conference on Machine Learning. PMLR, 2020
>
> [5] Li T, Sahu A K, Zaheer M, et al. Federated optimization in heterogeneous networks. Proceedings of Machine Learning and Systems, 2020
>
> [6] Zhang X, Hong M, Dhople S, et al. FedPD: A federated learning framework with adaptivity to non-IID data. IEEE Transactions on Signal Processing, 2021

---

> ### Author Response · Authors · 2022-11-07
> **Response to Reviewer W3wF (Part 1/3):**
>
> Thank you very much for your review and affirmation of this job. We'll answer your questions one by one in the following. We are also very honored to share some of our understandings with you.
>
> ## 1. About the question of "no improvement in terms of achieved rates":
> In the federated stochastic non-convex case, under the common assumptions of "smoothness", "bounded stochastic" and "heterogeneity", the $\mathcal{O}(\frac{1}{T})$ is a fast convergence rate. Compared with the previous works, our key contribution in the theoretical analysis is that we prove the prox-based methods could achieve $\mathcal{O}(\frac{1}{T})$ rate without relying on the strong assumption of bounded full gradient and the harsh requirement of the local minimal in each communication round.
>
> ## 2. About the question of "higher generalization accuracy in the title":
> We use "higher generalization accuracy" in the title. Thank you for pointing out this issue and we do not prove the generalization error bound theoretically. The core target of our paper focuses on the practical generalization accuracy of the real-world dataset, rather than the theoretical generalization error bound. We will add more introductions and explanations in our paper to avoid misunderstanding. We add the code demo files in the supplementary materials.
>
> In FL, it is difficult to strictly prove that an algorithm theoretically generalizes better than other methods. There is currently little work to analyze the stability of the iterative states $x_{t}$ to prove the strict generalization error bound, which requires more complex assumptions. Most of the works are based on the analysis of the additional constructed loss function, which is an approximation of the total optimization process. Here we provide an explanation for the improvement of generalization performance with the extra gradient steps.
>
> $\textbf{(What FedSpeed is doing with perturbation step?)}$ For vanilla SGD, the state $x_{t}$ is updated one step using the inverse gradient. The point it searches could be considered as a minimization of the loss function in the neighborhood of $x_{t}$. Conversely, if we take the ascent step at $x_{t}$, the loss function of the obtained point is maximized within $x_{t}$'s neighborhood. For FedSpeed, the algorithm first takes an ascent step to search for the point with a larger loss value within the neighborhood of $x_{t}$. Then it will calculate the extra gradient at this specific point with maximized loss value. This means that $x_{t}$ will update towards minimizing the loss function of itself and the loss-maximized point in its neighborhood.
>
> $\textbf{(FedSpeed can reach a flat minima.)}$ Vanilla SGD-type method only focus on the optimization of the $x_{t}$ itself. However, in the non-convex optimization, there are many equivalent local minimal. Using the extra ascent step of FedSpeed we can approach those local minimal with a flat landscape because as we stated before,  it consistently minimizes the loss function of the iterative point, and its neighbor loss-maximized point. This means that the algorithm can arrive at a point where the loss of itself, and that of its neighbor is minimized, which is a $\textbf{flat minima}$.
>
> $\textbf{(Flat minima guarantees a better generalization.)}$  The major difference between generalization and optimization is that they consider different data distributions (generalization always expects the true expectation and optimization usually considers the finite-sum problems in a finite number of data samples). Therefore, the true minima for the training and test dataset could be different. If the landscape of the obtained minima point is flat enough, the gaps between generalization and optimization will be much smaller. This is because a slight perturbation of the data distribution (resulting from the inconsistency of the testing and training distribution) would not result in an intense increase of loss for a flat minima point, and therefore ensure its generalization accuracy.

---

> ### Author Response · Authors · 2022-11-18
> **Due Reminder**
>
> Thank you for your review and questions in the first round！
> We want to know if our answers address your problems. The due is approaching, and we are unable to update the draft after the due.  If there are new questions we will answer and update the submission as soon as possible within the due.

---

> ### Author Response · Authors · 2022-12-08
> **Further discussion**
>
> Dear reviewer,
>
> Thanks for your valuable comments and suggestions. We have thoroughly addressed your concerns in the rebuttal and the revised manuscript. In addition, the source codes are also provided for better reproducibility. If you have new questions please let us know so that we can answer them as soon as possible within the due.
>
> Best, Authors

---

### Official Review · Reviewer_CRv7 · 2022-10-26

**Confidence:** 3
**Correctness:** 3
**Technical Novelty And Significance:** 3
**Empirical Novelty And Significance:** 3
**Recommendation:** 6

**Clarity, Quality, Novelty And Reproducibility:**

The paper is mostly clear. I appreciate the technical contribution of this paper, but it is hard for me to judge the originality since I'm not very familiar with federated learning.

**Strength And Weaknesses:**

The paper is well-written and easy to follow. The proposed algorithm achieves a convergence rate of O(1/T) by incorporating prox-correction terms and gradient perturbation techniques. I have not checked the proofs in the appendix, though major arguments seem to be correct.

My concern is that the experiments only compare the test accuracy. The paper may also present the training losses or gradient norms to verify the theoretical analysis.

**Summary Of The Paper:**

This paper studies federated learning for nonconvex optimization. The paper proposes a novel algorithm called FedSpeed, which applies a prox-correction term to reduce the bias and leverages gradient perturbation to improve the local generalization performance. FedSpeed achieves a fast convergence rate of O(1/T) and outperforms other baselines empirically.

**Summary Of The Review:**

Overall, I think this is an interesting paper with solid contributions to federated nonconvex optimization.

---

> ### Author Response · Authors · 2022-11-07
> **Response to Reviewer CRv7:**
>
> Thank you very much for your review and affirmation of our work. We'll answer your questions one by one in the following. We are also very honored to share some of our understandings with you.
>
> ## 1. About the question of "Federated Learning":
> Federated learning is an emerging distributed framework that jointly trains a global model with several edged devices. All the data is stored on the local client and can not be accessed by the other clients. They will communicate with the global server periodically to share the learning information. The global server generates the merged model without any local data, which can protect local data privacy. From the optimization view, researchers mainly focus on improving the performance via fewer communication rounds to save communication costs and reduce the client shift due to the heterogeneous dataset.
>
> ## 2. About the question of "the originality of FedSpeed in FL":
> First of all, we appreciate your review of this work.
>
> FedSpeed focuses on practical performance in the federated learning framework. We propose the correction term and the extra ascent gradient to improve the performance both in the theoretical analysis and empirical studies. The main contributions of this paper are i) theoretically, compared with the previous works, we provide a new analysis to prove that the prox-based method with the regularization term could achieve the fast $\mathcal{O}(\frac{1}{T})$ rate without the strong assumption of bounded full gradient and the harsh requirement of the local minimal per communication round. We provide the analysis on the local interval $K$ to indicate how to select the efficient $K$ and other hyperparameters, which could speed up the training. ii) we incorporate the correction term and gradient perturbation to further improve the practical performance. Extensive experiments are conducted to verify the efficiency of FedSpeed. To achieve the same accuracy, our proposed FedSpeed could adopt a larger local interval $K$ and reduce the communication round $T$ in the training than other baselines, which matches our theoretical analysis.
>
> ## 3. About the question of "the paper may also present the loss or gradient norms to verify the theoretical analysis":
> Our experiments section mainly refers to the experiment of the previous paper (baselines in our paper). Usually, researchers pay more attention to the performance of the test accuracy or its stability in practice in the FL. Therefore, we also omitted the report on the losses. We will add the figures in the revision. Here we report the communication rounds required to achieve the target loss value in the following Table. We add the code demo files in the supplementary materials. Thank you very much for your comments.
>
> | mehtod | loss=1.0 | loss=0.8 | loss=0.7 | loss=0.6 | loss=0.5 |
> |:--------------:|:---:|:----:|:----:|:-----:|:-----:|
> | FedAvg       | 63 | 184 |  -  |  -  |  -  |
> | FedAdam      | 66 | 189 | 488 |  -  |  -  |
> | FedCM        | 41 | 70  | 113 | 475 |  -  |
> | SCAFFOLD     | 60 | 106 | 169 | 294 |  -  |
> | FedDyn       | 61 | 73  | 118 | 209 |  -  |
> | FedSpeed     | 56 | 82  | 110 | 166 | 287 |
>
> Experiment setups: the same as the CIFAR-10 Dir-0.6 10%/100 in our paper.
> "-" means the algorithm can not achieve before 500 communication rounds.
>
> We report the loss value at the target communication round in the following Table.
> | mehtod | T=50 | T=100 | T=200 | T=500 |
> |:--------------:|:---:|:----:|:----:|:-----:|
> | FedAvg       | 1.26 | 0.90 | 0.78 | 0.71 |
> | FedAdam      | 1.07 | 0.89 | 0.77 | 0.69 |
> | FedCM        | 0.97 | 0.73 | 0.67 | 0.57 |
> | SCAFFOLD     | 1.12 | 0.83 | 0.66 | 0.56 |
> | FedDyn       | 1.08 | 0.77 | 0.61 | 0.52 |
> | FedSpeed     | 1.14 | 0.77 | 0.56 | 0.45 |
>
>
> #### It is a pleasure to discuss this with you, which will help us to further improve this work. We explain and prove the concerns mentioned in the reviews. If there are any questions, we are happy to continue the discussion with you. Thank you again for reading this rebuttal.

---

> ### Author Response · Authors · 2022-11-18
> **Due Reminder**
>
> Thank you for your review and questions in the first round！
> We want to know if our answers address your problems. The due is approaching, and we are unable to update the draft after the due.  If there are new questions we will answer and update the submission as soon as possible within the due.

---

> ### Author Response · Authors · 2022-12-08
> **Further discussion**
>
> Dear reviewer,
>
> Thanks for your valuable comments and suggestions. We have thoroughly addressed your concerns in the rebuttal and the revised manuscript. In addition, the source codes are also provided for better reproducibility. If you have new questions please let us know so that we can answer them as soon as possible within the due.
>
> Best, Authors

---

### Official Review · Reviewer_P9p1 · 2022-10-30

**Confidence:** 3
**Correctness:** 2
**Technical Novelty And Significance:** 2
**Empirical Novelty And Significance:** 2
**Recommendation:** 5

**Clarity, Quality, Novelty And Reproducibility:**

The presentation of this paper is clear and easy to follow in general. But there are quite a few typos and grammatical mistakes in this paper. The novelty of this paper is somewhat marginal since most of the proposed techniques exist in the literature and this paper tried to integrate these ingredients, although the integration is non-trivial. The reproducibility of this paper seems to be fine.

**Strength And Weaknesses:**

Strengths:
1. This paper considered how to mitigate data heterogeneity and the resultant client drift phenomenon, which is an important problem in FL.
2. The authors conducted a detailed theoretical convergence rate analysis
3. The authors conducted comprehensive experiments with existing FL algorithms.

Weaknesses:
1. Although the proposed FedSpeed algorithm contains some interesting ideas (momentum plus prox) and is shown to have a strong convergence rate performance, the FedSpeed algorithm has a large number of hyper-parameters: $\rho$, $\alpha$, $\eta$, and $\lambda$, which could be quite difficult to tune in practice. Also, in the experiments, although the authors mentioned to some degree how to choose these parameters, it's not entirely clear from the paper how these parameters are related to each other.

2. The $O(1/T)$ convergence rate in Theorem 4.4 seems questionable. Specifically, there is a constant $\kappa>0$, which necessitates a very small local learning rate $\eta_l$ and parameter $\rho$. Furthermore, $\eta_l$ depends on local steps $K=O(T)$. So in the end, the denominators of the first 3 terms on the right-hand side of Eq. (6) all have nontrivial dependence on $T$. Thus, it seems to claim $O(1/T)$ convergence rate is questionable (at least not holding without a careful justification).

3. Remark 4.7 also seems misleading. It is actually not safe to the performance will improve as $K$ increases. For the same reason as above, in Corollary 4.5 where the authors plugged in values to derive the dependence of $\Phi$ w.r.t. $\Phi$, the dependence of $\kappa$ on $T$ seems neglected. I would suggest the authors double-check all the calculations to make sure there is no error.

**Summary Of The Paper:**

This paper proposed a new federated learning (FL) algorithm called FedSpeed to mitigate the negative impacts of data heterogeneity and client drift problems in FL. The proposed method uses a prox-correction term on current local updates to reduce the bias introduced by the prox-term while retaining the benefits of the prox-term to maintain local consistency. The authors showed that FedSpeed has an $O(1/T)$ convergence rate to a constant error neighborhood and allows $O(T)$ local steps per communication round. The authors conducted experiments to compare FedSpeed with several existing FL algorithms.

**Summary Of The Review:**

This paper proposed a new algorithm called FedSpeed to mitigate the impact of data heterogeneity in FL and achieve a fast convergence rate under this setting. Although the problem is important and the proposed algorithm contains some interesting ideas, the algorithm has too many hyper-parameters to tune and could be hard to implement in practice. Perhaps also because of this reason, some of the convergence performance analysis seems questionable and may need more careful checking.

---

> ### Author Response · Authors · 2022-11-07
> **Response to Reviewer P9p1 (Part 2/2):**
>
> ## 3. About the question of "whether $\kappa$ could be regarded as a constant in Theorem 4.4":
> This proof technique is similar to [2] (details in its Appendix page.13 (a8) inequality and page.16 (b8) inequality) and is also widely used in some previous works. Here we prove the term $\frac{1}{\kappa}$ has a constant upper bound where $\kappa=\frac{1}{2}-3\alpha^{2}L^{2}\rho^{2}-1536\eta_{l}^{2}L^{2}K$.
>
> #### Firstly let $\alpha=0$ which means there is no ascent steps,
>
> $proof.$
>
> There exists a constant $c\in(0,\frac{1}{2})$, we let $\kappa=\frac{1}{2}-1536\eta_{l}^{2}L^{2}K \geq \frac{1}{2}-c > 0$,
>
> thus $\kappa=\frac{1}{2}-1536\eta_{l}^{2}L^{2}K \geq \frac{1}{2}-c$ when the $\eta_{l} \leq \frac{\sqrt{c}}{16\sqrt{6K}L} < \frac{1}{32\sqrt{3K}L}$.
>
> For the final convergence, it contains $\frac{1}{\kappa}$ terms, and $\frac{1}{\kappa}\leq \frac{2}{1-2c}$ which is a constant upper bound.
>
> #### Then we prove it when $\alpha\neq 0$,
>
> $proof.$
>
> There exists two constants $c_{1}\in(0,\frac{1}{2}), c_{2}\in(0,\frac{1}{2})$ and they satisfy $c_{1}+c_{2}\in(0,\frac{1}{2})$,
>
> we let $\frac{1}{2}-3\alpha^{2}L^{2}\rho^{2} > \frac{1}{2}-c_{1}$ and $\frac{1}{2}-1536\eta_{l}^{2}L^{2}K > \frac{1}{2}-c_{2}$, where the $\rho$ and $\eta_{l}$ satisfy $\rho < \frac{\sqrt{c_{1}}}{\sqrt{3}\alpha L} < \frac{1}{\sqrt{6}\alpha L}$ and $\eta_{l} < \frac{\sqrt{c_{2}}}{16\sqrt{6K}L} < \frac{1}{32\sqrt{3K}L}$.
>
> Then we can bound the $\kappa=\frac{1}{2}-3\alpha^{2}L^{2}\rho^{2}-1536\eta_{l}^{2}L^{2}K > \frac{1}{2}-c_{1}-c_{2} > 0$, and the term $\frac{1}{\kappa} < \frac{2}{1-2c_{1}-2c_{2}}$ which is a constant upper bound.
>
> Here, we discuss more about the final selection of $\rho$ and $\eta_{l}$. In the proof above, FedSpeed requires two additional contrains on $\rho < \frac{1}{\sqrt{6}\alpha L}$ and $\eta_{l} < \frac{1}{32\sqrt{3K}L}$. In our Corollary 4.5, considering these conditions, we let $\rho=\mathcal{O}(\frac{1}{\sqrt{T}})$ which satisfies the constant upper bound constrain and let $\eta_{l}=\mathcal{O}(\frac{1}{K})$ which satisfies the order of the upper bound $\mathcal{O}(\frac{1}{\sqrt{K}})$.
>
> In summary, though $\kappa$ contains the term of $\eta_{l}$ which is relevant to $T$, under this proof it can be considered as a constant bound. The key is that these two additional constrains won't break the selection of $\rho$ and $\eta_{l}$ to achieve the target convergence.
>
> [2] Achieving linear speedup with partial worker participation in non-IID federated learning. (ICLR 2021)
>
> ## 4. about the question of "$\Phi$ in Remark 4.7 do not consider the relationship of $\kappa$ and $T$ and it is not safe to the performance when $K$ increases":
> As proof in above, the $\kappa$ could be considered as a constant bound and this does not break the final convergence rate. Here, we discuss more. In fact, the order of $\kappa$ cannot be smaller than a constant bound for $\kappa=\frac{1}{2}-3\alpha^{2}L^{2}\rho^{2}-1536\eta_{l}^{2}L^{2}K$. The same, we let $\alpha=0$ for convergence, and $\kappa=\frac{1}{2}-1536\eta_{l}^{2}L^{2}K$. If $\kappa$ could approach a very small value, which requires the $1536\eta_{l}^{2}L^{2}K$ to approach $\frac{1}{2}$. This makes the learning rate $\eta_{l}$ to be fixed to approach $\frac{1}{32\sqrt{3K}L}$, which is unacceptable and impossible in the deep learning theory. According to the proof in answer.3 above, the convergence is correct in Theorem 4.4, Corollary 4.5, and Remark 4.7.
>
> Increasing $K$ to a larger value is proven in [3] under a strong assumption of the bounded full gradient. FedAvg can not achieve a larger $K$ to $\mathcal{O}(T)$ as we mentioned in Remark 4.7 under the common assumptions. While some prox-based methods like [4,5] can adopt a larger local interval because they require minimizing each sub-problem per communication round. However, the proofs of prox-based algorithms in federated learning are flawed. They all require the assumption of the exact local solution or the $\epsilon$-inexact solution per round. FedSpeed does not rely on the strong assumption of bounded full gradient and the requirement of the local minimal per communication round in the theoretical analysis and proves the efficiency of a larger $K$. Our experiments also show FedSpeed can adopt a larger interval $K$ to reduce the communication round $T$ with a small loss in precision.
>
>
> [3] Adaptive Federated Optimization (ICLR 2021)
>
> [4] Federated Learning Based on Dynamic Regularization (ICLR 2021)
>
> [5] FedADMM: A Robust Federated Deep Learning Framework with Adaptivity to System Heterogeneity
>
> #### It is a pleasure to discuss this with you, which will help us to further improve this work. We explain and prove the concerns mentioned in the reviews. If there are any questions, we are happy to continue the discussion with you. Thank you again for reading this rebuttal.

---

> ### Author Response · Authors · 2022-11-07
> **Response to Reviewer P9p1 (Part 1/2):**
>
> Thank you very much for your review and affirmation of our work. We'll answer your questions one by one in the following, including some misunderstandings and some essential academic questions worth exploring. We are also very honored to share some of our understandings with you.
>
> ## 1. About the question of "marginal novelty":
> Firstly, to the best of our knowledge, this is the first work to prove the prox-based methods could achieve the same convergence rate of $\mathcal{O}(\frac{1}{T})$ without relying on the strong assumption of bounded full gradient and the requirement of the local minimal per communication round. We discuss the values ​​of $K$ and $T$ and provide their theoretical ranges to achieve the target convergence.
>
> Secondly, we propose the efficient FedSpeed and conduct extensive experiments to test the properties of FedSpeed and verify its efficiency in the federated learning framework, especially in adopting a larger local interval $K$ to reduce communication rounds $T$ in practice. The test in Figure.2 of our paper shows that FedSpeed can achieve better test accuracy with fewer communication rounds on the real-world dataset. Also, as our answers in question 1., the selections of hyperparameters in FedSpeed are not difficult to tune, which is a truly practical method in FL.
>
> ## 2. About the question of "FedSpeed has a large number of hyper-parameters $\rho, \alpha, \eta_{l}, \lambda$ which could be quite difficult to tune in practice, and how these parameters are related to each other":
> Thank you for this question. Practicality is one of the necessary and important concerns in federated learning. In fact, $\eta_{l}$ and $\lambda$ are two basic hyperparameters in FL. FedSpeed proposes to use an extra ascent gradient step in the local training and adopts the $\rho$ and $\alpha$ to adjust it. We will detail the meanings and selections of these hyperparameters in this part.
>
> (1) $\eta_{l}$ is the learning rate which is a basic hyperparameter in deep learning, and usually, we do not finetune this for a fair comparison in the experiments. We just select the same common settings as the previous works mentioned.
>
> (2) $\lambda$ is the coefficient for the prox-term, which is proposed in the FedProx and a lot of prox-based federated methods adopt this hyperparameter widely both in personalized-FL and centralized-FL. The selection of this hyperparameter has been studied in many previous works which verify its efficiency. Usually, the selection of $\lambda$ is in $\{10, 100\}$ on the CIFAR-10/100 dataset, and we test it also works on the TinyImagenet.
>
> (3) $\rho$ is the ascent step learning rate. Like many extra gradient methods, the selection of $\rho$ is usually related to the local learning rate $\eta_{l}$. In order not to unduly affect the performance of the gradient descent, the learning rate for the extra gradient step $\rho$ is usually set not much larger than the learning rate for the gradient descent step $\eta_{l}$. Obviously, if $\rho$ is set very small, the updated state of the extra gradient steps will be very limited, which makes this operation have no effect. Therefore, the selection of $\rho$ usually matches that of $\eta_{l}$. In our experiments, the $\eta_{l}$ is set as $0.1$, which is a common selection in the previous works. We test the selection of $\rho$ in {$0, 0.01, 0.05, 0.1, 0.2$} which represents for {"no extra gradient", "$0.1\eta_{l}$", "$0.5\eta_{l}$, "$1\eta_{l}$", "$2\eta_{l}$"}. The best performing selection is $\rho=1\eta_{l}$ in CIFAR-10 (details in Section 5.3 paragraph "Learning rate $\rho$ for the gradient perturbation"). We also test this selection on the CIFAR-100 and TinyImagenet, and it also works well. We recommend that the selection of $\rho$ should be kept comparable to the learning rate $\eta_{l}$.
>
> (4) $\alpha$ is the ratio for merging the gradient of the extra ascent step. In FedSpeed, the $\alpha$ is in the range of $[0,1]$. The same, if $\alpha$ is set very small, which means it does not merge the gradient of the ascent steps. In our experiments, we test the selection of $\alpha$ in {$0, 0.5, 0.75, 0.875, 0.9375, 1.0$}. The best-performing selection is $\alpha=0.9375$ in CIFAR-10. In fact, $\alpha=1$ also works well (details in Section 5.3 paragraph "Perturbation weight $\alpha$"). Thus, about $\alpha$, we recommend that it should be close to $1.0$, e.g. for $0.9, 0.99, 1.0$. This also verifies the improvements of the ascent steps.\
>
> In summary, the powerful and efficient optimizer usually contains a large number of hyperparameters, e.g. for Adam (basic Adam contains $\eta_{l}$, first-order momentum weight $\beta_{1}$ and second-order momentum weight $\beta_{2}$). However, the hyperparameters in FedSpeed can be selected by simple experience as we mentioned above. In practice, FedSpeed ​​can achieve higher performance compared to other baselines without the complex tuning. We add the code demo files in the supplementary materials.

---

> ### Author Response · Authors · 2022-11-18
> **Due Reminder**
>
> Thank you for your review and questions in the first round！
> We want to know if our answers address your problems. The due is approaching, and we are unable to update the draft after the due.  If there are new questions we will answer and update the submission as soon as possible within the due.

---

> ### Author Response · Authors · 2022-12-08
> **Further Discussion**
>
> Dear reviewer,
>
> Thanks for your valuable comments and suggestions. We have thoroughly addressed your concerns in the rebuttal and the revised manuscript. In addition, the source codes are also provided for better reproducibility.  If you have new questions please let us know so that we can answer them as soon as possible within the due.
>
> Best,
> Authors

---

### Author Response · Authors · 2022-11-15
**Summary of Revisions**

## Summary of Revisions

First of all, we are very grateful to all reviewers for their reviews. We answer each reviewer's questions one by one earnestly. Although no reply has been received, we are still looking highly forward to receiving the feedback from reviewers to further improve this paper. Due to the deadline of the rebuttal is approaching, we submit the revision according to some problems mentioned by the reviewers in the first round (the newly added content is displayed in blue), which are summarized as follows:

(1) We add the code demo in the Supplementary Material, which is based on the Pytorch and the detailed usage can be referred in the readme.

(2) We add the introduction of the two mentioned references by the reviewer W3wF in the section of "Related Work" and discuss more in the Appendix.

(3) We add the loss curves in the experiments shown in the Appendix.

(4) We add the selection analysis of the hyperparameters in the Appendix to show that the hyperparameters are easy to tune in the practical training. Extensive experiments verify the efficiency of the FedSpeed in the real-world dataset.

(5) According to the question of the reviewer P9p1, we add the proof of "$\kappa$ have the constant bound" in the Appendix. Our convergence analysis is correct and can be strictly proved with respect to $\kappa$. In fact, this constant bound has been used widely in the previous works as we mentioned in the answers and citations.

In summary, in this paper, we propose a novel framework in the federated learning, which utilizes the correction term and the perturbation term, and could achieve the SOTA performance in the practical training. Our theoretical analysis relax the assumptions as mentioned by the reviewer W3wF to prove the FedSpeed method can greatly contribute to the convergence by adopting a larger local interval $K$ to reduce the communication rounds both theoretically and practically. It is a good property in the practical training to save the communication costs, which is also verified in the experiments. We provide the analysis to select the two hyperparameters $\alpha$ and $\rho$. Extensive ablation studies are conducted to show that they are easy to tune in practical training.

---

### Decision · Program_Chairs · 2023-01-20

**Decision:**

Accept: poster

**Justification For Why Not Higher Score:**

I believe there is limited scope in the work that a small subset of the ( ICLR ∩ FL ∩ OPT ∩ Theory ) will find interesting.

**Justification For Why Not Lower Score:**

There is significant theoretical novently

**Metareview: Summary, Strengths And Weaknesses:**

This paper presents FedSpeed, a novel and practical method for Federated Learning (FL) to alleviate the negative impacts posed by non-vanishing biases and local over-fitting. FedSpeed applies a prox-correction term to reduce the biases and merges the vanilla stochastic gradient with a perturbation computed from an extra gradient ascent step in the neighborhood. Theoretical analysis indicates that the convergence rate is related to both the communication rounds and local intervals with a tighter upper bound. Experiments on real-world dataset demonstrate that FedSpeed converges faster and achieves state-of-the-art performance compared to several other baselines.

Overall, the strengths of the paper include its focus on addressing data heterogeneity and client drift in federated learning, its theoretical analysis and comprehensive experiments, and the effectiveness of the proposed FedSpeed algorithm in achieving a fast convergence rate. However, there are also some weaknesses such as the large number of hyperparameters, the unclear explanation of how to choose these parameters, and potential issues with the convergence rate analysis and the claim that performance will improve with larger local updates. There may also be concerns about the originality of the paper and the lack of presentation of training losses or gradient norms in the experiments.

The authors have revised the paper in response to the reviewers' comments by adding the code demo in the supplementary material, discussing related references in the related work section, adding loss curves in the experiments and the selection analysis of hyperparameters in the appendix, and providing additional proofs. They convincingly argue that the paper has novelty in that it is the first to prove that prox-based methods can achieve the same convergence rate of O(1/T) without relying on strong assumptions and requirements, and that they have conducted extensive experiments to demonstrate the efficiency of their proposed FedSpeed algorithm in real-world datasets. They also address some misunderstandings and provide further explanations and justifications for their claims.

Overall, I believe this is a paper that is going to be interesting for the FL theory community, and is worth to be published.


**Note From Pc:**

if the above contains the word "oral" or "spotlight" please see: "oral" presentation means -> notable-top-5% and "spotlight" means -> notable-top-25%. As stated in our emails, we are disassociating presentation type from AC recommendations

**Summary Of Ac-Reviewer Meeting:**

N/A